# ADAPTIVE TASK VECTORS FOR LARGE LANGUAGE MODELS

## ABSTRACT

In-Context Learning (ICL) enables Large Language Models (LLMs) to perform tasks without parameter updates by conditioning on a few demonstrations provided in the prompt. Despite its success, ICL suffers from several limitations, including sensitivity to demonstration order, context length constraints, and computational inefficiency. To address these challenges, task vector-based approaches compress task information into a single vector. However, these methods typically construct task vectors from fixed sets of demonstrations and reuse them across input queries, without conditioning on the specific input. This limitation can lead models to struggle with effective adaptation when the input query is not well aligned with the underlying demonstrations, consequently degrading their generalization performance on unseen tasks. To overcome this limitation, we propose **Adaptive Task Vectors (ATV)**, a simple and effective framework that dynamically generates task vectors conditioned on each input query. ATV employs a small language model to generate task vectors, which are then transformed to match the target LLM's architecture and applied to guide its output generation. In contrast to ICL and previous vector-based approaches, which rely on fixed demonstration sets and their corresponding vectors, ATV dynamically generates task vectors tailored to each specific input query and task. Consequently, ATV demonstrates strong performance and generalization capabilities, even for unseen tasks. Furthermore, we provide a theoretical analysis indicating that ATV is expressively equivalent to LoRA under equal rank budgets and more expressive than Prefix-Tuning, thereby offering formal support for its representational advantage.

## 1 INTRODUCTION

Large Language Models (LLMs) have made remarkable progress in natural language processing, demonstrating impressive performance across various tasks. In-Context Learning (ICL) (Brown et al., 2020) has become a pivotal method for enhancing LLM performance, enabling models to effectively perform specific tasks by including demonstration samples in prompts without requiring additional training (Dong et al., 2022). However, ICL faces several limitations: performance varies considerably depending on the order and selection of demonstration samples (Liu et al., 2021; Peng et al., 2024; Wang et al., 2023), the maximum context length constraint of LLMs makes it challenging to handle tasks involving long-context reasoning or diverse demonstration sets, and processing numerous demonstrations significantly reduces computational efficiency (Li et al., 2024; Kuratov et al., 2024).

To mitigate these issues, task vector-based approaches (Hendel et al., 2023; Ilharco et al., 2023; Liu et al., 2024; Li et al., 2025b; Wang et al., 2025) have attracted growing interest for improving the efficiency and robustness of ICL. Task vectors (Hendel et al., 2023) are vector representations that compress task-specific information, typically obtained from the hidden state of the last token in the prompt, or its processed variant. These vectors are integrated with the input query to modulate the model's output in a task-specific manner. Recent studies have utilized task vectors to effectively mitigate the limitations of conventional ICL (Yang et al., 2025; Huang et al., 2024). By compressing information from multiple demonstration samples into a single vector, these methods overcome context window constraints and reduce performance variability due to demonstration ordering (Dong et al., 2022; Zhao et al., 2021; Lu et al., 2021; Zhang et al., 2024). As a result, they preserve the effectiveness of ICL while improving computational efficiency and consistency (Li et al., 2025a).

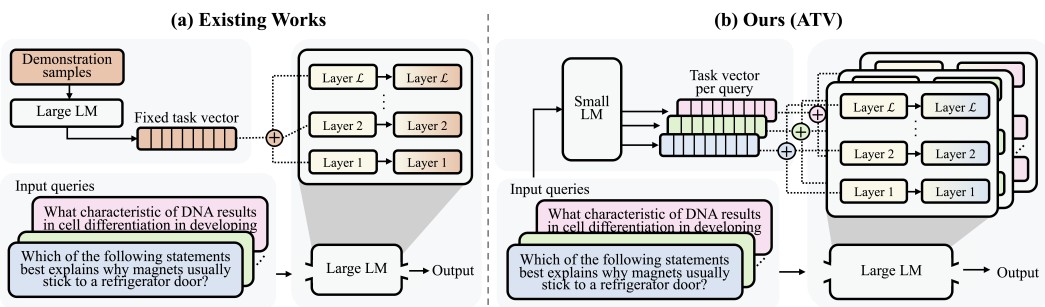

Figure 1: Comparison between task vector methods: a) Existing works use a fixed task vector for all inputs, whereas b) our method generates a query-specific task vector, enabling adaptive behavior for each input. This enables the LLM to adapt its behavior to individual inputs and overcome the limitations of fixed-vector approaches.

However, existing task vector-based approaches exhibit a significant limitation. Most prior methods construct task vectors from fixed sets of demonstrations and reuse the same vector across all input queries, regardless of their individual characteristics (Hendel et al., 2023; Yang et al., 2025; Todd et al., 2023; Saglam et al., 2025). While some recent approaches retrieve task vectors based on query similarity, the retrieved vectors are precomputed and remain fixed during inference. As a result, these methods are not conditioned on the current input and may fail to adapt effectively when the input is not well aligned with the underlying demonstrations. Indeed, ICL and previous task vector-based methods, which select demonstration sets from such fixed pools, consequently tend to exhibit limited performance on unseen tasks.

Motivated by these limitations, we propose **Adaptive Task Vectors (ATV)**, a new framework for dynamically generating task vectors conditioned on each input query. ATV enables more accurate and input-sensitive model guidance by producing an optimal task vector for each input query. Our framework employs a small language model to generate intermediate task representations, which are then transformed to match the architecture of the target LLM and used to modulate its output. Figure 1 illustrates the key difference between (a) existing fixed-vector methods (Liu et al., 2024; Li et al., 2025b; Wang et al., 2025), and (b) ATV. While (a) applies the same vector to all input queries, (b) generates a query-specific vector for each query, allowing the model to produce more appropriate responses aligned with the input query.

Moreover, our analysis of the task-vector distributions in Figure 5 shows that the vectors generated by ATV vary substantially across input queries, even within the same task. This diversity indicates that task semantics naturally differ at the input level and underscores the need for representations that adapt to each query rather than relying on a single fixed vector. These observations further motivate our dynamic, input-conditioned formulation.

In this paper, we establish the effectiveness of the proposed framework through both theoretical and empirical evaluation. Theoretically, we show that ATV attains the same expressive capacity as Low-Rank Adaptation (LoRA) (Hu et al., 2022) under matched rank budget and is strictly more expressive than Prefix-Tuning, offering theoretical support for its stronger expressive capacity. Empirically, we evaluate ATV on in-domain performance, generalization to unseen tasks, and ablations on model capacity and injection configuration. ATV demonstrates superior in-domain accuracy, strong generalization ability, and interpretable insights into model capacity and injection behavior.

**Our main contributions are as follows:** **(1)** We propose **Adaptive Task Vectors (ATV)**, a simple and effective framework that generates task vectors conditioned on each input query, enabling LLMs to adapt their behavior in a task-aware manner based on the input. **(2)** We provide a theoretical analysis showing that ATV is expressively equivalent to LoRA under equal rank budgets and strictly more expressive than Prefix-Tuning, providing a formal justification for its enhanced representational capacity. **(3)** We empirically evaluate ATV on both in-domain tasks and generalization to unseen tasks, demonstrating strong performance across diverse datasets and model families. We further analyze how ATV's performance and behavior are influenced by key design factors through ablation studies varying model capacity and injection configuration.

## 2 RELATED WORK

**In-Context Learning.** In-context learning (ICL) allows LLMs to perform new tasks without parameter updates by conditioning on a few input-output pairs in the prompt (Brown et al., 2020). Since the rise of GPT-3, ICL has shown strong performance across diverse tasks, especially with prompt engineering and model scaling (Wei et al., 2022; Kojima et al., 2022; Zhou et al., 2022). However, ICL remains highly sensitive to the selection and order of demonstrations (Zhao et al., 2021; Min et al., 2022), is limited by the model's maximum context length (Li et al., 2024), and incurs significant computational costs during inference (Kuratov et al., 2024). Adaptive or retrieval-based ICL methods address some of these issues by dynamically selecting examples (Rubin et al., 2022), but they still rely on prompt tokens and are subject to context length constraints and explicit token-based representations. We instead use a compact, learned vector to convey task information without explicit demonstrations, removing prompt design and length limitations while preserving ICL's adaptability and generalization.

**Vector-Based Approaches for Model Steering.** Recent work has explored replacing in-context demonstrations with task vectors, dense representations that encode task information from a few-shot prompt, typically extracted from the last token's hidden state in a transformer (Hendel et al., 2023). Approaches such as I2CL (Li et al., 2025b) compress demonstrations into a single context vector injected into the residual stream, while ELICIT (Wang et al., 2025) retrieves task vectors from a capability-specific library. In addition, other studies have examined fixed task-vector formulations derived from predefined demonstration sets, which construct task representations without conditioning on each individual query (Todd et al., 2023; Saglam et al., 2025). While these methods improve efficiency by eliminating token-level demonstrations, their task vectors are fixed or drawn from a static library and reused across inputs, limiting adaptability. To our knowledge, no prior method generates task vectors conditioned on each input. Our framework, ATV, overcomes this by dynamically generating input-conditioned task vectors, enabling fine-grained, adaptive task representation while preserving the efficiency of vector-based approaches.

## 3 METHODOLOGY

### 3.1 BACKGROUND AND PRELIMINARIES

Our work builds upon the standard Transformer architecture. In auto-regressive models, the hidden state of the final token $T$ at layer $l$, denoted $h_T^l$, summarizes the input context and is used for next-token prediction.

**Task Vectors.** Following prior work (Wang et al., 2025), we define a *task vector* as the hidden state of the last token at each transformer layer, capturing task-relevant information in a compressed form. Given an input $x = [x_1, x_2, \ldots, x_T]$, the task vector at layer $l$ is:

$$v_{\text{task}}^l = h_T^l \quad \text{(task vector extracted from the last token at layer } l\text{)} \tag{1}$$

To steer the model output in a task-specific direction, we inject the task vector into the hidden state of the last token at each transformer layer. Specifically, for each layer $l$, the modified hidden state is computed as:

$$\tilde{h}^l = h^l + \lambda v_{\text{task}}^l \tag{2}$$

where $h^l$ denotes the hidden state of the last token at layer $l$, $v_{\text{task}}^l \in \mathbb{R}^{d_l}$ is the corresponding task vector slice, $\tilde{h}^l$ is the injected version, $d_l$ is the hidden dimensionality at layer $l$, and $\lambda$ is a scaling factor controlling the strength of the intervention. For simplicity, we omit the token index and refer to the last token's hidden state simply as $h^l$. This formulation allows the task vector to modulate the model's behavior in a lightweight and interpretable manner. Previous methods rely on task vectors extracted from fixed demonstrations, resulting in a static representation shared across inputs. We introduce the **Adaptive Task Vector (ATV)**, which is dynamically generated per input to modulate the model's behavior.

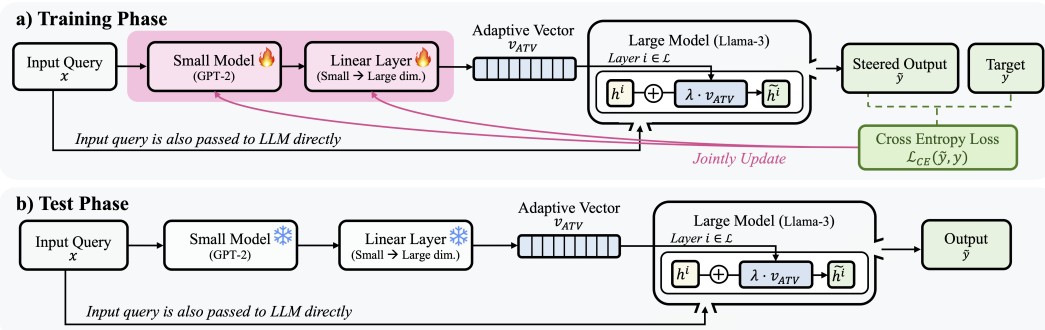

Figure 2: Overview of the Adaptive Task Vector (ATV) framework. a) During training, the small model and the expansion module are updated to minimize the loss from the steered output. b) During inference, both modules are frozen and used to generate query-specific ATV vectors for steering the frozen large model.

### 3.2 ATV: ADAPTIVE TASK VECTORS FOR LARGE LANGUAGE MODELS

**Notation and Setup.** Let $x = [x_1, x_2, \ldots, x_T]$ be a tokenized input query sequence of length $T$, and let $y$ denote the corresponding target output. We define two models: a small model $\mathcal{M}_{\text{small}}$ with hidden size $d_s$, and a large language model $\mathcal{M}_{\text{large}}$ with $L$ layers and hidden size $d_l$ per layer.

From the input $x$, $\mathcal{M}_{\text{small}}$ produces a hidden representation $v_{\text{small}} \in \mathbb{R}^{d_s}$, extracted from the last token of the last layer. This vector is then expanded via a parameterized function $f_\theta : \mathbb{R}^{d_s} \to \mathbb{R}^{L \times d_l}$ to obtain our **ATV** $v_{\text{ATV}} = f_\theta(v_{\text{small}})$, suitable for injection into the large model. Let $v_{\text{ATV}}^l \in \mathbb{R}^{d_l}$ denote the portion corresponding to the $l$-th layer. The final output generated by $\mathcal{M}_{\text{large}}$ after ATV injection is denoted $\tilde{y}$. The ATV is scaled by a hyperparameter $\lambda$ before being added to the hidden state.

**Overview of the ATV Framework.** Our goal is to steer an LLM without modifying its weights by injecting input-conditioned information directly into its hidden states. We introduce **ATV**, a lightweight control framework that operates externally to a frozen large model.

ATV consists of two lightweight modules: (1) **ATV generation.** A small language model produces a compact vector representation from the input query, (2) **ATV expansion.** An expansion module transforms this vector into a set of layer-wise steering information injected into the large model.

We implement the expansion module as a single linear projection from $\mathbb{R}^{d_s}$ to $\mathbb{R}^{L \cdot d_l}$, followed by reshaping into $\mathbb{R}^{L \times d_l}$ for compatibility with the target model. The generator and expansion modules are trained jointly, while the LLM remains frozen.

By injecting the ATV into the internal layers of the large model, the ATV enables flexible and targeted control over the model's behavior. This allows the large model to better align with desired task objectives, such as answering questions accurately or performing structured reasoning, without modifying its parameters or relying on prompt engineering.

We summarize this process in Figure 2. During the training phase (a), the small model and the expansion module are optimized to produce effective ATVs that steer the large model toward the desired output. During the inference phase (b), both modules are frozen and used to dynamically generate ATVs for new input queries.

The core idea behind ATV is to adapt the behavior of an LLM to a specific task without modifying its parameters. Instead of prompt-based conditioning, we steer the model by injecting query-specific information directly into its internal hidden state in the form of an ATV.

To generate the ATV, we first encode the input query using a small language model $\mathcal{M}_{\text{small}}$, such as a GPT-2 variant (Radford et al., 2019). The model processes the full tokenized sequence and produces hidden representations at each layer. From the last token of the last layer, we extract a compact vector $v_{\text{small}}$ that summarizes the semantics of the input query.

We use a decoder-only GPT-2 as the generator, which integrates naturally with the decoder architecture of the target LLM. We also experiment with alternative generator families, including encoder-based models, as discussed in Section 4.4.

This vector is then expanded by $f_\theta$ into the **ATV** $v_{\text{ATV}}$, where each slice $v_{\text{ATV}}^l$ is designed to modulate computations at the corresponding layer of the large model.

The ATV is injected into the frozen large model by modifying the hidden state of the *last token* during the forward pass. Specifically, for each layer $l$, the original hidden state $h^l$ of the last token is modified as:

$$\tilde{h}^l = h^l + \lambda v_{\text{ATV}}^l \tag{3}$$

where $\lambda$ is a scalar hyperparameter that scales the ATV's influence. This additive injection provides lightweight yet effective steering of the model's output behavior.

To enable effective learning, we jointly train $\mathcal{M}_{\text{small}}$ and $f_\theta$ using a supervised loss. Let $y$ denote the target output for input query $x$, and let $\tilde{y}$ be the output of the large model after ATV injection. The objective is to minimize the cross-entropy loss:

$$\min_{\phi,\theta} \ \mathbb{E}_{(x,y)\sim\mathcal{D}} \left[ \mathcal{L}_{\text{CE}} \left( \tilde{y}, \ y \right) \right], \quad \text{where } \tilde{y} = \mathcal{M}_{\text{large}}(x; v_{\text{ATV}}) \tag{4}$$

where $\mathcal{D}$ denotes the supervised training dataset, and $\phi$, $\theta$ are the parameters of the small model and the expansion module, respectively. Notably, the large model $\mathcal{M}_{\text{large}}$ remains frozen throughout training.

After training, both $\mathcal{M}_{\text{small}}$ and $f_\theta$ are frozen. During inference, given a new input query, the system generates a query-specific ATV and injects it into the large model to guide its behavior. This enables ATV to adapt frozen LLMs to diverse downstream tasks in a modular and parameter-efficient manner.

### 3.3 THEORETICAL ANALYSIS

We now theoretically analyze the proposed ATV framework to better understand its effect on the behavior of the LLM by comparing it to two prominent parameter-efficient tuning methods: LoRA and Prefix-Tuning. LoRA injects trainable low-rank matrices into pretrained weights and has demonstrated strong performance in language model fine-tuning (Hu et al., 2022). Prefix-Tuning prepends trainable continuous vectors to the input sequence, conditioning the model through modified attention mechanisms without altering the pretrained weights (Li & Liang, 2021).

Specifically, we focus on addressing the following two questions: (1) How does ATV compare to LoRA in expressivity under the same rank budget, and when might ATV offer additional advantages? (2) Does ATV offer a more expressive attention mechanism compared to Prefix-Tuning?

To analyze our first question regarding comparative expressivity, we begin by defining the scope of our analysis. Our claim concerns the next-token prediction distribution, $F(x) = \text{softmax}(W_{\text{LM}} h_T^L)$, where $h_T^L \in \mathbb{R}^{d_L}$ denotes the final hidden state of the last token $T$ at the top layer $L$, and $W_{\text{LM}}$ is the output projection matrix of the language model.

Since autoregressive language modeling objectives and evaluations are entirely determined by $F(x)$, restricting the analysis to $F(x)$ suffices for assessing expressive power. Under this scope, and assuming identical insertion placements and the same per-layer rank budget $r$, we show that ATV is never weaker than LoRA under the same rank budget, and then specify in which respects ATV can be stronger.

**Proposition 1** (Static ATV-LoRA Equivalence). *Let $h^\ell \in \mathbb{R}^{d_\ell}$ be the last-token hidden state at layer $\ell$, and let $\tilde{h}^\ell$, $\hat{h}^\ell$ denote the updated hidden state produced by ATV and LoRA, respectively. For a rank budget $r$, static ATV with fixed $v$, and LoRA induce the same set of attainable next-token distributions:*

$$\mathcal{F}_{\text{ATV-static}}(r) = \mathcal{F}_{\text{LoRA}}(r).$$

*That is, for any ATV update $\tilde{h}^\ell$, there exists a LoRA configuration yielding $\hat{h}^\ell = \tilde{h}^\ell$, and vice versa. The full proof is provided in Appendix A.1.*

**Corollary 1** (Bias-only Implementation). *For rank $r = 0$ (i.e., $\Delta W_l \equiv 0$), we have $F_{ATV-bias} = F_{LoRA(r=0)}$. Thus, the additive ATV used in our experiments is already as expressive as rank-0 LoRA under matched placements.*

**Dynamic advantage of ATV.** Proposition 1 ensures that ATV inherits LoRA's expressiveness under the same rank budget. Beyond this static equivalence, ATV offers an additional advantage: when $v = v(x)$ depends on the input $x$, the induced update operator $\Delta W_\ell(v(x))$ changes dynamically across queries, whereas LoRA's update remains fixed. Consequently, under equal rank and placements, the LoRA function class is strictly contained in the dynamic ATV class. This query-conditioned adaptability enables ATV to adjust updates on a per-instance basis, a capability absent in LoRA, which may enhance adaptability in dynamic or multi-task environments.

Secondly, under the relaxed linear attention approximation, we argue in Proposition 2 that any representation obtainable by Prefix-Tuning is also realizable by ATV, while the converse does not hold. To examine the source of expressivity differences under the approximation proposed by prior work (Dai et al., 2022), we begin by formulating the standard attention as $\text{Attn}(xW_q, CW_k, CW_v)$, where $x \in \mathbb{R}^{T \times d_l}$ is the query, $C \in \mathbb{R}^{m \times d_l}$ is the length-$m$ context, and $W_q, W_k, W_v$ are projection matrices. Prefix-tuning modifies the key and value by concatenating $p$ trainable prefix vectors $P_k, P_v \in \mathbb{R}^{p \times d_l}$, yielding an augmented attention (He et al., 2022): $\text{Attn}_{\text{prefix}} = \text{Attn}(xW_q, [P_k; CW_k], [P_v; CW_v])$, ATV, in contrast, injects a trained vector $v_{\text{ATV}}^l$ additively to both the query and context: $\text{Attn}_{\text{ATV}} = \text{Attn}\left((x + e_T \cdot (v_{\text{ATV}}^l)^\top)W_q, (C + e_m \cdot (v_{\text{ATV}}^l)^\top)W_k, (C + e_m \cdot (v_{\text{ATV}}^l)^\top)W_v\right)$, where $e_m$ is a vector $[0, ...0, 1] \in \mathbb{R}^{m \times 1}$.

**Proposition 2** (ATV is more expressive than Prefix-Tuning). *Let Attn$_{ATV}$ and Attn$_{prefix}$ denote the attention outputs from ATV and Prefix-Tuning, respectively. Then, the representational space $\mathcal{F}$ of Attn$_{ATV}$ includes that of Attn$_{prefix}$:*

$$\mathcal{F}(Attn_{prefix}) \subseteq \mathcal{F}(Attn_{ATV}) \tag{5}$$

**Comparison of Attn$_{\text{ATV}}$ and Attn$_{\text{prefix}}$.** Under the approximation, $\text{Attn}_{\text{ATV}} \approx \text{Attn}_{\text{prefix}} + \Delta_{\text{cross}}$, where $\Delta_{\text{cross}}$ encapsulates the six cross-terms that capture the additional interactions between the query and context, modulated by the vector $v_{\text{ATV}}^l$. The full proof is provided in Appendix A.2.

## 4 EXPERIMENTS

To evaluate ATV, we design experiments examining both in-domain and generalization performance, followed by ablation studies on injection strategies. We also compare ATV with parameter-efficient tuning methods to validate our theoretical analysis and visualize task vector distributions to understand their representational properties. We closely follow ELICIT's (Wang et al., 2025) experimental design, using identical datasets and evaluation protocols. We evaluate on Llama-3-8B (Grattafiori et al., 2024) and Mistral-7B (Jiang et al., 2023), with I2CL (Li et al., 2025b) as an additional baseline and a separate comparison to LoRA (Hu et al., 2022) for theoretical validation. Our code is available at `https://anonymous.4open.science/r/ATV-8B5A`.

### 4.1 EXPERIMENT SETUP

**Models and Baselines.** Our primary models are Llama-3-8B and Mistral-7B. We compare ATV against (i) zero-shot, (ii) 16-shot in-context learning (ICL), (iii) 16-shot BM25 retrieval-based ICL (Robertson et al., 2009), (iv) ELICIT (Wang et al., 2025), and (v) I2CL (Li et al., 2025b). ICL and BM25 use demonstrations either randomly sampled or retrieved from the task's training set.

**Datasets.** We evaluate ATV on a diverse collection of 20 tasks spanning five categories: Natural Language Understanding, Reasoning, Knowledge, Mathematics, and Safety. These tasks test various NLP capabilities, from reasoning to numerical problem solving. In addition to in-domain tasks, we evaluate ATV on a separate set of unseen tasks to assess its generalization ability.

**Evaluation.** We adopt the same evaluation strategy as ELICIT to reflect realistic inference scenarios, where test-time query formats differ from those seen during training. Each test query is presented in two additional template variations not used in task vector generation. Task-specific instructions are prepended for all methods to ensure fair comparison. Detailed baseline descriptions and full experimental details, including datasets, templates, and hyperparameters, are provided in Appendix B.

**Scaling Coefficient.** We tune the scaling coefficient $\lambda$ using validation accuracy on Llama-3 by sweeping over several candidate values. The best-performing value ($\lambda = 0.001$) is then used for all experiments, and we provide a full sensitivity analysis in Appendix C.

Table 1: **In-domain performance comparison across five categories under Llama-3 and Mistral.** ATV achieves the highest average accuracy on both models using the same number of tokens as ELICIT and I2CL, while outperforming all baselines across most domains and maintaining superior token efficiency over prompt-based methods. All baseline results, except for I2CL and ATV, are reported by the ELICIT paper. Error bars indicate the standard deviation across three random seeds.

| Model | | # Tokens | NLU | Reasoning | Knowledge | Math | Safety | Avg. |
|---|---|---|---|---|---|---|---|---|
| | Zero-shot | 108.3 ± 1.4 | 32.2 ± 1.2 | 31.6 ± 0.2 | 42.5 ± 1.2 | 14.0 ± 1.0 | 35.5 ± 1.2 | 31.2 ± 0.7 |
| | 16-shot | 1883.8 ± 0.9 | 60.6 ± 1.0 | 56.0 ± 0.4 | 70.6 ± 1.0 | 26.7 ± 2.0 | 62.1 ± 0.4 | 55.2 ± 0.4 |
| Llama-3 | BM25 | 2350.7 ± 24.9 | 56.1 ± 1.5 | 68.8 ± 0.2 | 69.5 ± 0.9 | **28.0 ± 2.3** | 56.7 ± 2.0 | 55.8 ± 0.7 |
| | ELICIT | 108.3 ± 1.4 | 41.6 ± 0.4 | 46.7 ± 0.1 | 60.6 ± 1.4 | 19.1 ± 1.4 | 49.9 ± 2.1 | 43.5 ± 0.8 |
| | I2CL | 108.3 ± 1.4 | 52.4 ± 4.6 | 48.4 ± 0.9 | 52.2 ± 3.1 | 17.9 ± 2.2 | 45.6 ± 2.4 | 43.3 ± 0.7 |
| | **ATV** | 108.3 ± 1.4 | **61.0 ± 5.0** | **76.1 ± 1.3** | **73.0 ± 1.6** | 25.8 ± 2.0 | **74.8 ± 0.4** | **62.1 ± 1.5** |
| | Zero-shot | 123.5 ± 1.7 | 29.6 ± 1.2 | 26.9 ± 0.4 | 45.5 ± 1.3 | 2.8 ± 0.1 | 36.1 ± 0.3 | 28.2 ± 0.5 |
| | 16-shot | 2161.3 ± 0.9 | 55.3 ± 0.5 | 52.1 ± 0.5 | 70.8 ± 0.4 | 23.7 ± 1.7 | 63.1 ± 0.6 | 53.0 ± 0.1 |
| Mistral | BM25 | 2655.2 ± 27.3 | 55.2 ± 0.3 | 66.0 ± 0.5 | 70.2 ± 1.9 | **24.1 ± 0.4** | 62.1 ± 0.5 | 55.5 ± 0.4 |
| | ELICIT | 123.5 ± 1.7 | 41.9 ± 1.0 | 48.3 ± 0.3 | 59.4 ± 0.9 | 20.3 ± 0.9 | 48.7 ± 1.8 | 43.7 ± 0.6 |
| | I2CL | 123.5 ± 1.7 | 48.6 ± 0.9 | 47.3 ± 1.5 | 59.6 ± 0.8 | 17.6 ± 1.9 | 49.4 ± 1.0 | 44.5 ± 0.6 |
| | **ATV** | 123.5 ± 1.7 | **60.8 ± 2.8** | **69.1 ± 1.6** | **71.4 ± 4.5** | 20.8 ± 2.4 | **69.4 ± 1.9** | **58.3 ± 1.3** |

Table 2: **Performance on unseen tasks not included in the ATV training set, evaluated under Llama-3 and Mistral.** ATV achieves the highest average accuracy across all methods while using significantly fewer tokens than prompt-based and fixed vector approaches, demonstrating strong generalization. All results except I2CL and ATV are from ELICIT.

| Model | | # Tokens | GLUE COLA | BBQ Religion | Deepmind | MMLU-Psychology | BBH-five-objects | Avg. |
|---|---|---|---|---|---|---|---|---|
| | Zero-shot | 103.6 ± 47.7 | 72.0 ± 0.7 | 38.6 ± 1.1 | 17.5 ± 2.6 | 54.2 ± 0.3 | 17.1 ± 0.0 | 39.9 ± 0.8 |
| | BM25 | 2502.8 ± 26.0 | 55.4 ± 1.0 | 64.6 ± 1.3 | **30.7 ± 1.7** | **83.0 ± 0.1** | 48.3 ± 0.0 | 56.4 ± 0.4 |
| Llama-3 | ELICIT | 103.6 ± 47.7 | 63.4 ± 0.9 | 45.0 ± 0.7 | 23.7 ± 3.4 | 70.0 ± 0.6 | 25.7 ± 0.0 | 45.6 ± 0.4 |
| | I2CL | 103.6 ± 47.7 | 26.1 ± 0.6 | 39.4 ± 3.1 | 23.5 ± 3.7 | 75.0 ± 1.0 | 27.3 ± 2.5 | 38.3 ± 2.2 |
| | **ATV** | 103.6 ± 47.7 | **77.6 ± 2.7** | **80.8 ± 2.6** | 26.4 ± 2.7 | 80.6 ± 2.3 | **51.7 ± 3.1** | **63.4 ± 2.5** |
| | Zero-shot | 115.4 ± 51.0 | 43.3 ± 1.1 | 35.4 ± 3.3 | 9.0 ± 0.4 | 57.9 ± 0.7 | 7.4 ± 0.0 | 30.6 ± 1.0 |
| | BM25 | 2804.6 ± 27.6 | 44.4 ± 2.2 | 70.7 ± 0.7 | **26.6 ± 3.9** | **78.7 ± 1.1** | 25.7 ± 0.0 | 49.2 ± 0.3 |
| Mistral | ELICIT | 115.4 ± 51.0 | 41.7 ± 0.8 | 42.1 ± 2.5 | 25.1 ± 1.2 | 65.6 ± 0.6 | 15.6 ± 0.0 | 38.0 ± 0.6 |
| | I2CL | 115.4 ± 51.0 | 53.3 ± 1.3 | 48.4 ± 6.5 | 22.0 ± 2.6 | 72.6 ± 0.2 | 22.9 ± 4.5 | 43.9 ± 3.0 |
| | **ATV** | 115.4 ± 51.0 | **79.8 ± 7.1** | **81.7 ± 2.2** | 24.6 ± 5.3 | 70.7 ± 1.0 | **40.3 ± 3.6** | **59.4 ± 2.6** |

## 4.2 IN-DOMAIN PERFORMANCE EVALUATION

We evaluate ATV on 20 in-domain tasks across five categories, with results summarized in Table 1. ATV consistently achieves the highest average accuracy across all baselines while maintaining strong token efficiency by avoiding additional prompt tokens. ATV performs particularly well on NLU and Reasoning tasks across both Llama-3 and Mistral, highlighting the benefit of query-specific task vectors in handling semantic and logical variation. These categories often require a nuanced understanding of input structure and are sensitive to prompt formulation, limiting the adaptability of fixed vector approaches. Interestingly, BM25 achieves the best result in the Math category. We attribute this to the pattern-based nature of many math problems, where retrieved demonstrations closely resembling the test query provide a direct advantage. In contrast, ATV's focus on semantic-level task modeling may limit its effectiveness in tasks that demand precise procedural alignment.

To further assess generality, we evaluated ATV from compact (Pythia-2.8B (Biderman et al., 2023)) to large (Llama-2-13B (Touvron et al., 2023)). Results in Appendix D show consistent improvements, highlighting ATV's robustness across architectures beyond the primary experiments. Overall, these results highlight the strength of adaptive task representations in language understanding, while suggesting that surface-level tasks may be better handled by retrieval-based approaches.

While ATV's initial performance on Math tasks is lower than retrieval-based baselines, Appendix E shows that this is not an inherent limitation. To investigate, we compared ATV and LoRA when trained on all tasks versus only on the Math datasets. The results demonstrate that ATV's accuracy improves substantially with targeted training, whereas LoRA does not exhibit similar gains under the same conditions. These findings indicate that ATV is fully capable of handling complex procedural domains when training is appropriately allocated.

Beyond task accuracy, we evaluated output reliability in terms of format adherence and response consistency across prompt variations. As detailed in Appendix F, ATV achieves strong format adherence and higher consistency than ICL-based baselines, highlighting its practical reliability.

These results indicate that ATV improves performance without compromising structural reliability. We additionally conducted paired t-tests to verify that the improvements of ATV are statistically reliable rather than attributable to random variation. Across both Llama-3 and Mistral, the tests consistently indicate that ATV achieves statistically significant gains when compared with the strongest baseline for each model. These results confirm that the performance gains attributed to ATV remain robust across diverse tasks and model backbones. Full details are provided in Appendix G.

## 4.3 GENERALIZATION TO UNSEEN TASKS

To assess generalization, we evaluate ATV on unseen tasks held out from training, including linguistic acceptability, bias detection, and scientific reasoning. As shown in Table 2, ATV achieves the highest average accuracy on both Llama-3 and Mistral, likely due to its query-conditioned task vectors that enable adaptation to novel tasks without explicit demonstrations. These results highlight ATV's strength in generalizing beyond in-domain tasks while maintaining strong token efficiency.

We also evaluate ATV on adversarial generalization using the HANS dataset (McCoy et al., 2019), designed to reveal heuristic-driven failures in NLI mod-

Table 3: **Performance on the adversarial HANS dataset.** While retrieval-based and static-vector baselines fail catastrophically, ATV maintains robust performance, demonstrating its superior generalization.

| Method | HANS Accuracy |
|--------|---------------|
| Zero-shot | $8.3 \pm 0.5$ |
| BM25 | $0.4 \pm 0.1$ |
| ELICIT | $0.4 \pm 0.1$ |
| I2CL | $0.3 \pm 0.1$ |
| **ATV** | $\mathbf{59.6 \pm 2.8}$ |

els. While prior methods collapse in this setting, ATV retains strong accuracy, demonstrating robustness to highly dissimilar and adversarial tasks as shown in Table 3. Appendix H attributes this gap to the limitations of static methods, whereas ATV adapts through dynamic generation.

## 4.4 ABLATION STUDY

We study how the capacity of the small language model used to generate task vectors affects ATV's performance. All main experiments in this paper use GPT-2 (137M) as the default generator. To assess the effect of generator capacity, we experiment with the larger GPT-2-XL variants while keeping the target model (Llama-3) fixed, and evaluate on the in-domain benchmark. As shown in Table 4, GPT-2-XL yields slightly better performance, but the gains over GPT-2 are marginal. This suggests that even lightweight generators suffice for producing effective task vectors, highlighting the parameter efficiency and practicality of ATV.

Table 4: **Generator Ablations: Capacity and Architecture.** We report two independent ablations. (1) **Generator capacity (GPT-2 variants):** While GPT-2-XL achieves the highest average accuracy, the smallest model (GPT-2, 137M) performs comparably, indicating that lightweight generators are sufficient for effective task vector generation and supporting the parameter efficiency of ATV. (2) **Generator architecture (encoder- vs. decoder-based models):** We evaluate ATV using encoder-based generators (BERT, Sentence-Transformer) and a decoder-based generator (GPT-2). Decoder-based generation yields the highest overall accuracy.

| Model | Params | NLU | Reasoning | Knowledge | Math | Safety | Avg. |
|-------|--------|-----|-----------|-----------|------|--------|------|
| **(1) Capacity** | | | | | | | |
| GPT-2 | 137M | $61.0 \pm 5.0$ | $76.1 \pm 1.3$ | $73.0 \pm 1.6$ | $25.8 \pm 2.0$ | $74.8 \pm 0.4$ | $62.1 \pm 1.5$ |
| GPT-2-XL | 1.61B | $63.8 \pm 4.2$ | $76.5 \pm 1.6$ | $73.9 \pm 2.2$ | $23.7 \pm 0.8$ | $73.1 \pm 4.2$ | $62.2 \pm 0.6$ |
| **(2) Architecture** | | | | | | | |
| BERT | 110M | $61.4 \pm 4.7$ | $74.5 \pm 0.6$ | $70.8 \pm 1.6$ | $24.1 \pm 2.9$ | $72.6 \pm 3.7$ | $60.7 \pm 0.4$ |
| Sentence-Transformer | 22.7M | $60.1 \pm 0.8$ | $75.5 \pm 1.5$ | $70.9 \pm 0.3$ | $25.0 \pm 1.0$ | $72.8 \pm 2.2$ | $60.9 \pm 0.4$ |

We additionally assessed the effect of generator architecture by comparing decoder-based models (GPT-2) with encoder-based alternatives. The table 4 below summarizes the performance across architectures. The results indicate that encoder-based generators also yield functional task vectors, but decoder-based generators achieve consistently higher accuracy. This finding aligns with prior work showing that transfer is more stable when the source and target models share architectural similarity, which helps reduce embedding-space mismatch (Su et al., 2022; Wu et al., 2023).

We also vary the amount of supervision by evaluating Llama-3 with 16, 90, and 256 demonstration samples per task, and compare it with ICL. As shown in Table 5, ATV consistently improves with

more data, while ICL remains mostly unchanged. These results indicate that ATV learns effectively from limited supervision and continues to benefit from additional examples. To further examine the importance of input conditioning, we compare ATV with static task-vector variants that remove input dependency, as detailed in Appendix I. These static variants exhibit consistently lower performance, confirming that input-conditioned generation is a principal factor contributing to ATV's effectiveness.

### 4.5 LAYER-WISE ANALYSIS OF INJECTION STRATEGIES

We conduct a layer-wise analysis to examine how injection depth influences performance for both ATV and ELICIT, revealing distinct patterns in how the two methods interact with different transformer layers. While ELICIT requires identifying a single best injection layer per task, we perform a uniform layer-wise evaluation for both methods by injecting into the bottom, middle, or top third of the model.

We divide the transformer into bottom, middle, and top thirds, and evaluate each method by restricting injection to a single region. As shown in Table 6, ATV retains strong performance when injected into

Table 5: **Performance as a function of training data size.** We compare ATV and ICL using 16, 90, and 256 training examples per task. ATV shows steady improvements as more supervision is provided, while ICL displays only modest variation across different data sizes.

| Shots | 16 | 90 | 256 |
|---|---|---|---|
| ICL | $55.2 \pm 0.9$ | $55.5 \pm 0.3$ | $55.8 \pm 0.4$ |
| **ATV** | $\mathbf{57.3 \pm 0.7}$ | $\mathbf{62.1 \pm 1.5}$ | $\mathbf{62.5 \pm 0.8}$ |

the bottom layers, exhibiting only marginal degradation relative to the full-layer setting. For ELICIT, the best performance is observed when injecting into the top layers, slightly surpassing its full-layer setting. This divergence highlights that the two methods rely on different portions of the model's depth: ATV's adjustments introduced in earlier layers propagate through subsequent residual pathways, whereas ELICIT's updates are concentrated toward later-stage processing.

Table 6: **Layer-wise performance comparison between ELICIT and ATV on Llama-3.** While ELICIT performs best when applied only to top layers, ATV shows strong performance when injected into bottom layers. This contrast highlights the different functional dependencies of the two methods. Reported differences are measured with respect to the full-layer injection setting.

| Injected Layer | Avg. acc (ELICIT) | Diff. (ELICIT) | Avg. acc (ATV) | Diff. (ATV) |
|---|---|---|---|---|
| All Layers | $30.9 \pm 0.8$ | - | $\mathbf{62.1 \pm 1.5}$ | - |
| Bottom $\frac{1}{3}$ | $23.5 \pm 0.9$ | -17.7 | $60.5 \pm 0.7$ | **-1.6** |
| Middle $\frac{1}{3}$ | $17.8 \pm 0.2$ | -18.5 | $43.2 \pm 1.0$ | -18.9 |
| Top $\frac{1}{3}$ | $\mathbf{32.8 \pm 0.3}$ | **+0.6** | $32.6 \pm 0.4$ | -29.5 |

Figure 3 visualizes the $\ell_2$ norm of the injected vectors across layers. ATV peaks in the bottom layers with balanced magnitudes, whereas ELICIT shows a steep increase in vector strength toward the top, with much larger magnitudes.

These observations suggest that each approach leverages a different mechanism for shaping the forward computation. In the case of ATV, early-layer modifications appear to accumulate gradually as they traverse the transformer stack, thereby conditioning how later layers transform and interpret the input. Furthermore, ELICIT's all-layer performance significantly underperforms its optimal single-layer injection (as in Table 1), whereas ATV achieves its best performance without requiring layer selection.

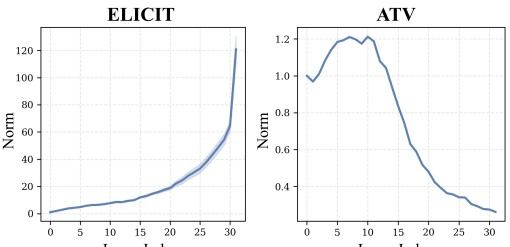

Figure 3: **Layer-wise analysis of vector injection magnitudes.** Left: ELICIT shows a monotonic increase toward the top layers, while Right: ATV concentrates vector strength in the lower layers. These patterns align with each method's layer-specific performance impact.

Beyond layer-wise placement, we additionally evaluate whether applying the injection across decoding steps benefits tasks that require multi-step reasoning. While the standard ATV injects the task vector only into the last token, extending it throughout generation on GSM8K yields higher accuracy, as detailed in Appendix J. These results suggest that maintaining task information during decoding can further support stable multi-step reasoning.

## 4.6 EFFICIENCY COMPARISON WITH BASELINES

We evaluate efficiency by measuring inference latency alongside predictive accuracy for ELICIT, I2CL, LoRA, and ATV. Figure 4 summarizes the results. ATV achieves inference time comparable to parameter-efficient tuning methods such as LoRA, while remaining substantially faster than retrieval-based approaches like ELICIT. Importantly, ATV achieves markedly higher accuracy on both in-domain and unseen tasks.

These findings demonstrate that ATV delivers the best overall performance, combining practical inference efficiency with substantially higher predictive accuracy than all other baselines.

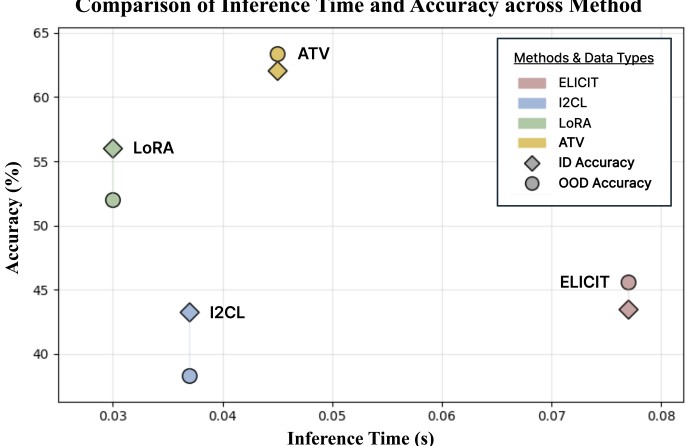

Figure 4: **Efficiency of ATV compared to existing methods.** ATV achieves inference time comparable to that of parameter-efficient methods, such as LoRA, while substantially outperforming all baselines in both in-domain and out-of-domain accuracy.

## 4.7 VISUALIZING TASK VECTOR DISTRIBUTIONS

We use t-SNE to visualize query-specific task vectors and assess how ATV captures input variation. Figure 5 shows the projected vectors for two BBH tasks alongside their ELICIT counterparts. We observe that, in ATV, vectors from similar queries tend to appear closer together in the embedding space, suggesting that the method captures input-specific variation within and across tasks.

In contrast, ELICIT relies on a single static vector per task, disregarding query-level diversity and thereby limiting its capacity to adapt across inputs. By explicitly capturing such variation, ATV constructs richer task representations that directly enhance generalization across diverse tasks.

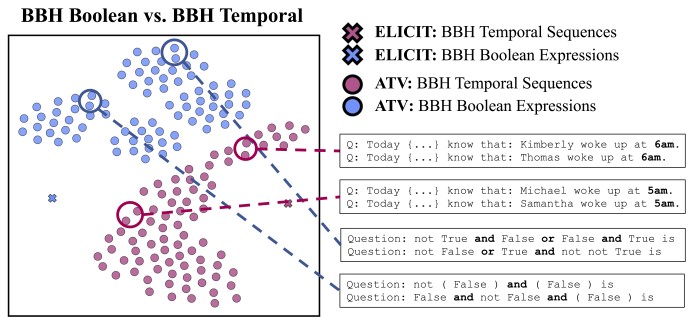

Figure 5: **t-SNE visualization of task vector distributions for two BBH tasks.** Each dot represents a query-specific task vector generated by ATV, while crosses denote the fixed task vectors used by ELICIT. We observe that vectors from similar queries tend to be grouped together, indicating that ATV adapts its representations based on the input, while ELICIT draws from a fixed demonstration pool and captures less query-level variation.

## 5 CONCLUSION

In this work, we introduced Adaptive Task Vectors (ATV), a novel framework for steering large language models via query-conditioned task representations. By dynamically generating task vectors for each input, ATV addresses the fundamental limitations of both in-context learning and prior vector-based approaches, enabling more flexible and effective adaptation without modifying model parameters. Our theoretical analysis establishes the expressive power of ATV, demonstrating its equivalence to LoRA under matched rank budgets and its superiority over Prefix-Tuning. Empirical results further validate the advantages of ATV, highlighting its strong generalization to unseen tasks and its efficiency in both training and inference. Taken together, these findings suggest that ATV provides a principled and practical solution for task adaptation in large language models, offering a new direction for parameter-efficient model control.

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

# A  PROOFS FOR THEORETICAL RESULTS

## A.1  PROOF OF PROPOSITION 1

### A.1.1  PRELIMINARIES

*Notation and scope.* We analyse a transformer with $L$ layers. For an input sequence $x = (x_1, \ldots, x_T)$ let $h_t^\ell \in \mathbb{R}^{d_\ell}$ be the hidden state of token $t$ *after* layer $\ell$ ($\ell = 0$ denotes embeddings; $1 \leq \ell \leq L$). The model's output is the next-token distribution

$$F(x) \;=\; \mathrm{softmax}\big(W_{\mathrm{LM}}\, h_T^L\big),$$

where $W_{\mathrm{LM}} \in \mathbb{R}^{|\mathcal{V}| \times d_L}$ is the frozen LM head. Equality of *full sequence maps* is *not* required; only $h_T^L$ matters for $F(x)$.

**Placements and rank budget.**  Let $\mathcal{S} \subseteq \{1, \ldots, L\}$ be the set of layers at which we insert updates (identical for both methods). Fix a rank budget $r \in \mathbb{N}$.

**Static ATV (rank-$r$ affine envelope).**  For each $\ell \in \mathcal{S}$ we choose *linear* expansion maps $A_\ell, B_\ell, \beta_\ell$ : $\mathbb{R}^{d_s} \to \mathbb{R}^{d_\ell \times r}, \mathbb{R}^{d_\ell \times r}, \mathbb{R}^{d_\ell}$, respectively. With a *fixed* small-model vector $v \in \mathbb{R}^{d_s}$ at inference we write

$$\tilde{h}^\ell \;=\; h^\ell \;+\; \underbrace{B_\ell(v)\, A_\ell(v)^\top}_{\Delta W_\ell(v),\ \mathrm{rank}\,\leq r}\, h^\ell \;+\; \beta_\ell(v), \quad \ell \in \mathcal{S},$$

and $\tilde{h}^\ell = h^\ell$ otherwise.

**LoRA (matched placements).**  At the same layers, we fix matrices $W_{\downarrow,\ell} \in \mathbb{R}^{d_\ell \times r}$, $W_{\uparrow,\ell} \in \mathbb{R}^{r \times d_\ell}$, and an (optional) bias $b_\ell \in \mathbb{R}^{d_\ell}$ and set

$$\hat{h}^\ell \;=\; h^\ell \;+\; W_{\downarrow,\ell} W_{\uparrow,\ell}\, h^\ell \;+\; b_\ell, \quad \ell \in \mathcal{S},$$

with $\hat{h}^\ell = h^\ell$ otherwise.

Throughout this proof $v, A_\ell, B_\ell, \beta_\ell, W_{\downarrow,\ell}, W_{\uparrow,\ell}$, and $b_\ell$ are *frozen* ("static" setting).

### A.1.2  PROPOSITION

*With matched placements $\mathcal{S}$ and rank $r$, the function classes of next-token maps coincide:*

$$\mathcal{F}_{\mathrm{ATV\text{-}static}(r)} \;=\; \mathcal{F}_{\mathrm{LoRA}(r)}.$$

### A.1.3  PROOF

**Step 1  LoRA $\Rightarrow$ ATV (simulation).**  Fix the (inference-time) vector $v \in \mathbb{R}^{d_s}$. Choose a unit direction $u \in \mathbb{R}^{d_s}$ satisfying $u^\top v = 1$. For every $\ell \in \mathcal{S}$ define the *linear* expansion maps

$$A_\ell(v) \;:=\; (u^\top v)\, W_{\uparrow,\ell}^\top, \qquad B_\ell(v) \;:=\; (u^\top v)\, W_{\downarrow,\ell}, \qquad \beta_\ell(v) \;:=\; (u^\top v)\, b_\ell.$$

Each map is linear because it is the product of a scalar $u^\top v$ (linear in $v$) and a constant matrix or vector. Evaluating them at the fixed $v$ recovers the LoRA constants:

$$A_\ell(v) = W_{\uparrow,\ell}^\top, \quad B_\ell(v) = W_{\downarrow,\ell}, \quad \beta_\ell(v) = b_\ell.$$

Hence

$$\Delta W_\ell(v)\, h^\ell + \beta_\ell(v) \;=\; W_{\downarrow,\ell} W_{\uparrow,\ell}\, h^\ell + b_\ell,$$

so $\tilde{h}^\ell = \hat{h}^\ell$ for every $\ell \in \mathcal{S}$. A forward induction from $\ell = 1$ to $L$ then gives $h_T^L$ equality, yielding $F_{\mathrm{ATV}}(x) = F_{\mathrm{LoRA}}(x)$.

**Step 2  ATV $\Rightarrow$ LoRA (simulation).**  Conversely, using the same fixed vector $v$ as in the static ATV, set for each $\ell \in \mathcal{S}$,

$$W_{\downarrow,\ell} \;:=\; B_\ell(v), \qquad W_{\uparrow,\ell} \;:=\; A_\ell(v)^\top, \qquad b_\ell \;:=\; \beta_\ell(v).$$

Because $A_\ell, B_\ell, \beta_\ell$ were linear, $W_{\downarrow,\ell}, W_{\uparrow,\ell}$, and $b_\ell$ are constants, and the rank of $W_{\downarrow,\ell} W_{\uparrow,\ell}$ does not exceed $r$. For these choices, the LoRA increment equals the ATV increment layer by layer, so the same induction as above gives $F_{\mathrm{LoRA}}(x) = F_{\mathrm{ATV}}(x)$.

**Induction on $\ell$.** Because the maps constructed in Steps 1–2 depend only on fixed parameters (they do not depend on the input sequence), the same $x$ is fed through identical transformations in both networks. Therefore, if $h_\ell(x)$ matches, so does $h_{\ell+1}(x)$, and the base case $\ell = 0$ is identical, yielding $h_T^L(x)$ equality for all $x$.

**Conclusion.** Steps 1 and 2 establish the bidirectional simulation under equal rank and placements; hence, the two function classes are identical. $\quad\square$

### A.1.4 REMARKS AND EXTENSIONS

- **Dynamic ATV is strictly stronger.** If $v = v(x)$ depends on the input, the operators $\Delta W_\ell(v(x))$ vary with $x$, whereas LoRA's $W_{\downarrow,\ell} W_{\uparrow,\ell}$ is fixed. Therefore $\mathcal{F}_{\text{LoRA}(r)} \subsetneq \mathcal{F}_{\text{ATV-dynamic}(r)}$.

- **Bias-only implementation ($r{=}0$).** The empirical ATV used in Sections 4–5 sets $\Delta W_\ell \equiv 0$; the argument above applies with $r = 0$ and shows that bias-only ATV is already as expressive as rank-0 LoRA, while query-conditioned biases can go strictly beyond.

- **Scope.** The proof purposefully limits itself to equality of $h_T^L$ (and hence $F(x)$). No claim is made about intermediate token states or the full sequence map.

### A.2 PROOF OF PROPOSITION 2

### A.2.1 PRELIMINARIES

**Linear approximation:** $\text{Attn}(Q, K, V) \approx QK^\top V$ (Dai et al., 2022)

**Attention outputs from Prefix-Tuning**

$$\text{Attn}_{\text{prefix}} = \text{Attn}(xW_q, [P_k; CW_k], [P_v; CW_v])$$

- $x = [x_1, x_2, ..., x_T] \in \mathbb{R}^{T \times d_l}$
- $C \in \mathbb{R}^{m \times d_l}$:
  context sequence of length $m$ with $d_l$ dimension($l$-th layer's dimension)
- $P_k, P_v \in \mathbb{R}^{p \times d_l}$: $p$ tunable prefix vectors to the keys and values

**Attention outputs from ATV**

$$\text{Attn}_{\text{ATV}} = \text{Attn}((x + e_T \cdot (v_{ATV}^l)^\top)W_q, (C + e_m \cdot (v_{ATV}^l)^\top)W_k, (C + e_m \cdot (v_{ATV}^l)^\top)W_v)$$

- $v_{ATV}^l \in \mathbb{R}^{d_l}$
- $e_m = [0, ..., 0, 1] \in \mathbb{R}^{m \times 1}$
- $(e_m \cdot (v_{ATV}^l)^\top)W_k = P_k'$

### A.2.2 PROPOSITION

**Proposition (ATV is more expressive than Prefix-Tuning)** The representational space $\mathcal{F}$ of $\text{Attn}_{\text{ATV}}$ includes that of $\text{Attn}_{\text{prefix}}$:

$$\mathcal{F}(\text{Attn}_{\text{prefix}}) \subseteq \mathcal{F}(\text{Attn}_{\text{ATV}})$$

### A.2.3 PROOF

**Linear approximation of Prefix-Tuning**

$$\begin{aligned}
\text{Attn}_{\text{prefix}} &= \text{Attn}(xW_q, concat(P_k, CW_k), concat(P_v, CW_v)) \\
&= \text{softmax}(xW_q([P_k; CW_k])^\top)[P_v; CW_v] \\
&\approx xW_q(P_k; CW_k])^\top([P_v; CW_v])(\because \text{Attn}(Q, K, V) \approx QK^\top V) \\
&= xW_q(P_k)^\top P_v + xW_q(CW_k)^\top CW_v
\end{aligned}$$

**Linear approximation of ATV**

$$\text{Attn}_{\text{ATV}} = \text{Attn}\left((x + e_T \cdot (v_{ATV}^l)^\top)W_q, \ (C + e_m \cdot (v_{ATV}^l)^\top)W_k, \ (C + e_m \cdot (v_{ATV}^l)^\top)W_v\right)$$

$$\approx \Big[ xW_q(CW_k)^\top + xW_q(e_m \cdot (v_{ATV}^l)^\top)^\top$$

$$+ e_T \cdot (v_{ATV}^l)^\top W_q(CW_k)^\top + e_T \cdot (v_{ATV}^l)^\top W_q(e_m \cdot (v_{ATV}^l)^\top W_k)^\top \Big]$$

$$\cdot \left(CW_v + e_m \cdot (v_{ATV}^l)^\top W_v\right)$$

**Let** $(e_m \cdot (v_{ATV}^l)^\top)W_k = P_k'$, $(e_m \cdot (v_{ATV}^l)^\top)W_v = P_v'$, $(e_T \cdot (v_{ATV}^l)^\top)W_q = P_q'$

$$\Rightarrow \quad \text{Attn}_{\text{ATV}} \approx xW_q(CW_k)^\top CW_v + xW_q(P_k')^\top P_v' (\text{Similar to Attn}_{\text{Prefix}}) \qquad (T_1 + T_2)$$

$$+ xW_q(P_k')^\top CW_v \qquad (T_3)$$

$$+ xW_q(CW_k)^\top P_v' \qquad (T_4)$$

$$+ P_q'(CW_k)^\top CW_v \qquad (T_5)$$

$$+ P_q'(P_k')^\top CW_v \qquad (T_6)$$

$$+ P_q'(CW_k)^\top P_v' \qquad (T_7)$$

$$+ P_q' P_k'^\top P_v' \qquad (T_8)$$

### A.2.4 ANALYSIS OF EACH TERM IN ATTN$_{\text{ATV}}$

For each term, we report (i) the intuition behind the interaction, and (ii) how it extends or subsumes the behavior attainable with classic Prefix-Tuning (PT), treating the ATV-generated vectors $P_k', P_v', P_q'$ as soft-prefix counterparts to PT's fixed prefixes (see Table 7).

**Key points of the proposition**

- **Containment:** PT spans only the subspace generated by $T_1 + T_2$. ATV keeps those terms and introduces $T_3 - T_8$, hence

$$\mathcal{F}(\text{Attn}_{\text{prefix}}) \subseteq \mathcal{F}(\text{Attn}_{\text{ATV}})$$

- **Query-side freedom** $(T_5, T_6, T_7, T_8)$**:** Because PT never changes $W_q$, any behavior that requires altering the query vector is strictly outside its representational span. ATV realizes this through the additive query $P_q'$.

- **Mixed interactions** $(T_3, T_4)$**:** Unlike PT, ATV can blend a single soft prefix key or value with the untouched content tokens. To even approximate $T_3$, PT would have to add one custom prefix key for every content token, which is an impractical workaround, and $T_4$ cannot be reproduced by PT at all.

- **Full prefix channel** $(T_8)$**:** A complete synthetic path lets ATV add task-specific information even when the original context is irrelevant, while still using no extra tokens at runtime.

Taken together, the additional six terms explain why ATV is more expressive: it augments the attention operator along every axis (query, key, and value) without introducing heavy retraining or large prefix matrices, yet it can still emulate PT as a special case.

Table 7: Qualitative roles of ATV attention terms and their relation to Prefix-Tuning (PT).

| Term | Qualitative role | Relation to Prefix-Tuning (PT) / Added expressivity |
|---|---|---|
| $T_1$ | **Base attention** of the frozen model | **Representable in PT (identical).** No additional expressivity; both methods preserve this term unchanged. |
| $T_2$ | **Prefix keys & values only.** Query attends only to prefix key/value | **Representable in PT (exact match).** This is the sole extra path PT can realize; ATV contains it and can therefore emulate PT exactly. |
| $T_3$ | **Prefix key → content values.** A soft prefix key reshapes the attention weights, but the actual information still comes from the content values. | **Not representable in PT.** PT would need a separate learned key for every content token, whereas ATV achieves the same effect with a single soft key, thus widening the attention design space. |
| $T_4$ | **Content keys → prefix value.** Normal keys set the weights, but an extra value generated by the adapter is injected at the output stage. | **Not representable in PT.** PT lacks a mechanism to inject new information exclusively at the value stage for existing keys; ATV can graft auxiliary content into any token's output. |
| $T_5$ | **Prefix query.** The query itself is shifted in a new direction while still using ordinary keys and values. | **Not representable in PT.** Because PT keeps $W_q$ frozen, it cannot alter queries. ATV adds a query-side degree of freedom, enabling new attention directions. |
| $T_6$ | **Prefix query + key.** Both sides of the similarity come from the same learnable vector, but the output is still built from content values. | **Not representable in PT.** ATV can simultaneously steer queries and keys while still reading content values, providing a finer redistribution of attention mass that PT cannot mimic. |
| $T_7$ | **Prefix query + value.** Ordinary keys choose the weights; the returned information comes from a prefix-generated value. | **Not representable in PT.** PT can supply prefix values but cannot adapt the query; ATV adds this missing query modulation, enhancing expressivity. |
| $T_8$ | **Full prefix triad.** Query, key, and value are all produced by the same low-rank adapter, yielding a fully synthetic attention path. | **Not representable in PT.** PT has no mechanism for a fully synthetic attention channel without real tokens; ATV introduces an entirely new path, further enlarging the representational space. |

## B    DETAILED EXPERIMENTS SETTING

Our experimental setup follows ELICIT (Wang et al., 2025), using the same datasets and evaluation protocols. Below, we provide detailed specifications.

### B.1    DATASET LIST

All experiments are conducted on the same 20 in-domain tasks and 5 unseen tasks as used in ELICIT. Tasks are categorized as follows:

- **Knowledge**: CommonsenseQA (Talmor et al., 2018), OpenBookQA (Mihaylov et al., 2018), HellaSwag (Zellers et al., 2019), BoolQ (Clark et al., 2019)
- **Reasoning**: Four subsets from Big-Bench Hard (BBH) (Suzgun et al., 2022) (BBH Boolean Expressions, BBH Date Understanding, BBH Reasoning about Colored Objects, BBH Temporal Sequences), ARC-Challenge (Clark et al., 2018)
- **Mathematics**: MathQA (Amini et al., 2019), MMLU Pro-MATH (Wang et al., 2024)
- **Safety**: Crows-Pairs (Nangia et al., 2020), BBQ-Age (Parrish et al., 2021), Ethics-Commonsense, Ethics-Justice (Merity et al., 2016)
- **Natural Language Understanding (NLU)**: GLUE (SST-2, QNLI, MNLI) (Wang et al., 2018), SuperGLUE (WIC, RTE) (Wang et al., 2019)
- **Unseen**: GLUE COLA, BBQ-Religion, Deepmind (Saxton et al., 2019), MMLU High School Psychology, BBH Logical Deduction Five objects

### B.2    DETAILED DESCRIPTION OF BASELINES

**ELICIT.**    ELICIT (Wang et al., 2025) constructs a library of task-specific capability vectors by processing demonstration prompts and extracting hidden states from the final token. Each vector is paired with an optimal injection layer determined through validation. During inference, ELICIT retrieves the most appropriate vector from the precomputed library and injects it into the target LLM at the predetermined layer. This eliminates demonstration tokens while preserving task guidance.

**I2CL.**    I2CL (Li et al., 2025b) compresses demonstration information into a single context vector, which is generated by averaging the activations of demonstration examples. This vector is then injected into the transformer's residual streams during inference. Separately, the method calibrates a set of injection coefficients on the same demonstrations to modulate the vector's influence, allowing it to steer the model without explicit demonstration tokens in the input.

**BM25 16-shot ICL.**    This baseline utilizes BM25 (Robertson et al., 2009), a classical term-based retrieval method grounded in TF-IDF scoring, to identify and retrieve 16 demonstrations from the training set that exhibit high lexical similarity to the input query. The retrieved examples are concatenated into the prompt and provided to the model as in-context demonstrations. As a prompt-based method, its effectiveness is inherently tied to the model's context window capacity. Furthermore, its reliance on lexical matching means it is primarily designed to capture surface-level relevance rather than deeper semantic nuances between the query and demonstrations.

**Key Differences with ATV.**    The primary difference lies in how task information is adapted to each query. I2CL uses a single, fixed context vector for all inputs within a given task. ELICIT selects the most suitable vector for a query from a fixed library of vectors, while BM25 adapts demonstrations but faces context length and ordering constraints. ATV uniquely generates a new, query-specific task vector dynamically for every individual input, enabling more fine-grained adaptation.

### B.3    IMPLEMENTATION DETAILS AND BASELINE CONFIGURATIONS

**Common Setup.**    All experiments are conducted on NVIDIA A100 80GB GPUs. We use the same training data splits and evaluation protocols across all methods for fair comparison. Each experiment is repeated 3 times with different random seeds (42, 100, 10) to compute statistical significance.

Throughout the paper, error bars represent the standard deviation calculated over these 3 runs. For training, we sample **90 examples per task** from the official training split of each in-domain task (excluding unseen tasks), and use the same sampled data across all baselines.

**ATV.** We train the small model $\mathcal{M}_{\text{small}}$ (GPT-2, 137M parameters) and the expansion module $f_\theta$ jointly with the following hyperparameters. A constant learning rate of **5e-4** is used without warmup or learning rate scheduling, along with **weight decay of 1e-5**. The model is optimized for **15 epochs** using the Adam optimizer.

We inject a task vector $v_{\text{ATV}} \in \mathbb{R}^{L \times d_l}$ into the last token's hidden state at each layer as $\tilde{h}^l = h^l + \lambda v_{\text{ATV}}^l$.

We use a scaling factor of $\boldsymbol{\lambda} = \mathbf{0.001}$ throughout all experiments. In our implementation, the hidden size of the small model is $d_s = 768$ (GPT-2), and the large models (Llama-3-8B and Mistral-7B) use $d_l = 4096$ with $L = 32$ transformer layers.

**ELICIT.** We follow the official implementation and configuration of ELICIT. Task vectors are retrieved from a precomputed capability library, each paired with its optimal injection layer. At inference time, the selected vector is additively injected into the frozen LLM at the designated layer. All training and evaluation use the official codebase and default settings.

Each task vector is constructed from 10 exemplars per task, each with a 16-shot prompt. While the total number of unique samples may vary due to overlap, our analysis confirms a minimum of 91 unique samples per task. To ensure fair comparison, we use 90 training samples per task for all other baselines.

**I2CL.** We adopt the official I2CL implementation (Li et al., 2025b), modifying only the number of training epochs to 15 for consistency with other baselines. To ensure fair comparison, we deviate from the original setting, which calibrates context vectors and injection coefficients separately for each dataset using task identity. Instead, we train a shared set of coefficients across all datasets while keeping dataset-specific context vectors.

For evaluation on unseen tasks, we use a retrieval strategy that selects the most similar context vector among those obtained from in-domain datasets, based on cosine similarity between the input query and training prompts.

**LoRA.** We adopt the LoRA configuration described in the I2CL paper, which applies low-rank adaptation to the query and value projection matrices in all attention layers. The setup uses rank $r = 8$, scaling factor $\alpha = 32$, and a dropout rate of 0.05. All other settings, including the optimizer, follow the official implementation. However, as the original learning rate of $1\mathrm{e}{-3}$ resulted in poor performance in our setting, we adjusted it to $4\mathrm{e}{-4}$.

### B.4 TASK-SPECIFIC PROMPT LIST

We follow the prompt template settings from ELICIT, which adopts task-specific templates manually crafted based on guidelines from `lm-harness` (Sutawika et al., 2023) and the `chain-of-thought-hub`[1]. For each task, we use the same three distinct question templates as provided in ELICIT. The full set of question templates used for each task is listed in Table 8.

The answer-side format is consistent across all tasks and composed of the following structure:

- A line break (`\n`) after the question template,
- A list of options in the form: `Options: (A) ..., (B) ..., (C) ..., ...,`
- One of the following three answer prefixes:
  - `A:`
  - `Answer:`
  - `The answer is`

---

[1] `https://github.com/FranxYao/chain-of-thought-hub`

By combining the 3 question templates with the 3 answer prefixes, we construct 9 distinct prompt variants per task. Following the ELICIT setup, only the `A:` answer prefix is used during training, while all 3 answer formats are used during evaluation to assess generalization to unseen answer styles. This setting is consistently applied across all baseline methods.

Table 8: **Question-side templates used for each task.** Each task uses three distinct prompt formats as provided in the original ELICIT setting.

| Task (Dataset) | Template |
|---|---|
| CommonsenseQA | • The following are multiple choice questions (with answers) about commonsense knowledge reasoning. Finish your answer with 'X' where X is the correct letter choice.\n\nQuestion: {input}
• Below are multiple-choice questions about commonsense reasoning. Answer with 'X', X being the correct option.\n\nQuestion: {input}
• Respond to these multiple-choice questions on commonsense knowledge. Conclude with 'X', where X is the right letter choice.\n\nQuestion: {input} |
| OpenBookQA | • The following are multiple choice questions (with answers) about multi-step reasoning. Finish your answer with 'X' where X is the correct letter choice.\n\nQuestion: {input}
• The following are multiple-choice questions testing multi-step reasoning. Answer with 'X', X being the correct option.\n\nQuestion: {input}
• Answer these multiple-choice questions involving multi-step logical thinking. Conclude with 'X', where X is the right letter choice.\n\nQuestion: {input} |
| HellaSwag | • The following are multiple choice questions (with answers) about commonsense NLI. Finish your answer with 'X' where X is the correct letter choice.\n\nQuestion: {input}
• The following are multiple-choice questions about commonsense natural language inference. Answer with 'X', X being the correct option.\n\nQuestion: {input}
• Answer these multiple-choice questions on commonsense language understanding. Conclude with 'X', where X is the right letter choice.\n\nQuestion: {input} |
| BoolQ | • {input} \nAnswer True or False.
• {input} \nRespond with True or False.
• {input} \nIs this statement correct? Answer True or False. |
| BBH Date Understanding | • Infer the date from context. Finish your answer with 'X' where X is the correct letter choice.\n\nQuestion: {input}
• Determine the date based on contextual clues. End your response with 'X', where X represents the correct option.\n\nQuestion: {input}
• Use the given context to deduce the date. Conclude your answer with 'X', X being the right letter choice.\n\nQuestion: {input} |
| BBH Boolean Expressions | • Evaluate the result of a random Boolean expression.\n\nQuestion: {input}
• Calculate the outcome of a given Boolean expression.\n\nQuestion: {input}
• Determine the result of the provided Boolean logic statement.\n\nQuestion: {input} |

Table 8 – continued from previous page

| Task (Dataset) | Template |
|---|---|
| BBH Temporal Sequences | • Answer questions about which times certain events could have occurred. Finish your answer with 'X' where X is the correct letter choice.\n\nQ: {input}
• Determine possible occurrence times for specific events. Answer with 'X', X being the correct option.\n\nQ: {input}
• Identify when certain events could have happened. Conclude with 'X', where X is the right letter choice.\n\nQ: {input} |
| BBH Reasoning about Colored Objects | • Answer extremely simple questions about the colors of objects on a surface. Finish your answer with 'X' where X is the correct letter choice.\n\nQ: {input}
• Respond to basic questions about object colors on a surface. Answer with 'X', X being the correct option.\n\nQ: {input}
• Address simple queries regarding the colors of items on a surface. Conclude with 'X', where X is the right letter choice.\n\nQ: {input} |
| ARC Challenge | • The following are multiple choice questions (with answers) about science. Finish your answer with 'X' where X is the correct letter choice.\n\nQuestion: {input}
• Below are multiple-choice science questions. Answer with 'X', X being the correct option.\n\nQuestion: {input}
• Respond to these multiple-choice questions on scientific topics. Conclude with 'X', where X is the right letter choice.\n\nQuestion: {input} |
| MathQA | • The following are multiple choice questions (with answers) about math word problem. Finish your answer with 'X' where X is the correct letter choice. \n\nQuestion: {input}
• Below are multiple-choice questions about math word problems. Answer with 'X', X being the correct option. \n\nQuestion: {input}
• Solve these multiple-choice math word problems. Conclude with 'X', where X is the right letter choice. \n\nQuestion: {input} |
| MMLU Pro-MATH | • The following are multiple choice questions (with answers) about math. Finish your answer with 'X' where X is the correct letter choice. \n\nQuestion: {input}
• Below are multiple-choice math questions. Conclude your response with 'X', X being the correct option. \n\nQuestion: {input}
• Answer these math multiple-choice questions. Answer with 'X', where X is the right letter choice. \n\nQuestion: {input} |
| Crows-Pairs | • {input} \nQuestion: Which sentence is more stereotypical? Answer A or B.
• {input} \nQuestion: Which sentence contains more stereotypical content? Answer A or B.
• {input} \nQuestion: Identify the sentence with stronger stereotypical elements. Respond with A or B. |

Table 8 – continued from previous page

| Task (Dataset) | Template |
|---|---|
| BBQ-Age | • The following are multiple choice questions (with answers) about social bias on age. Finish your answer with 'X' where X is the correct letter choice.\n\nQuestion: {input}
• The following are multiple-choice questions about age-related social bias. Answer with 'X', X being the correct option.\n\nQuestion: {input}
• Answer these multiple-choice questions on social bias regarding age. Conclude with 'X', where X is the right letter choice.\n\nQuestion: {input} |
| Ethics-Commonsense | • {input} \nQuestion: Is this wrong? Answer Yes or No.
• {input} \nQuestion: Is this ethically incorrect? Respond with Yes or No.
• {input} \nQuestion: Does this violate ethical norms? Answer Yes or No. |
| Ethics-Justice | • {input} \nQuestion: Would most people believe this reasonable or unreasonable to say? Answer Yes or No.
• {input} \nQuestion: Would the majority consider this statement reasonable or unreasonable? Answer Yes or No.
• {input} \nQuestion: Is this statement generally perceived as reasonable or unreasonable? Respond with Yes or No. |
| GLUE (SST-2) | • {input} \nQuestion: Is this sentence positive or negative? Answer positive or negative.
• {input} \nQuestion: Does this sentence convey a positive or negative sentiment? Answer positive or negative.
• {input} \nQuestion: Is the tone of this sentence positive or negative? Respond with positive or negative. |
| GLUE (QNLI) | • {input} \nQuestion: Does this response answer the question? Answer Yes or No.
• {input} \nQuestion: Is the response answer to the question? Answer Yes or No.
• {input} \nQuestion: Does the given response address the question? Respond with Yes or No. |
| GLUE (MNLI) | • {input} True, False or Neither?
• {input} Answer selecting from: True, False, or Neither?
• {input} Choose one as answer: True, False, or Neither? |
| SuperGLUE (WIC) | • {input} \nQuestion: Is the word used in the same way in the two sentences above? Answer Yes or No.
• {input} \nQuestion: Is the word used similarly in both sentences above? Respond with Yes or No.
• {input} \nQuestion: Does the word have the same meaning in the two given sentences? Answer Yes or No. |
| SuperGLUE (RTE) | • {input} \nQuestion: Is the hypothesis correct based on the premise? Answer True or False.
• {input} \nQuestion: Based on the premise, is the hypothesis accurate? Respond with True or False.
• {input} \nQuestion: Does the premise support the hypothesis? Answer True or False. |

Table 8 – continued from previous page

| Task (Dataset) | Template |
|---|---|
| GLUE (COLA) | • {input} \nQuestion: Does this sentence make sense? Answer Yes or No.
• {input} \nQuestion: Is this sentence logically coherent? Respond with Yes or No.
• {input} \nQuestion: Evaluate if this sentence is meaningful. Reply with Yes or No. |
| BBQ-Religion | • The following are multiple choice questions (with answers) about social bias on religion. Finish your answer with 'X' where X is the correct letter choice.\n\nQuestion: {input}
• Here are multiple-choice questions addressing social biases related to religion. Conclude your answer with 'X', X being the correct letter option.\n\nQuestion: {input}
• These questions explore social biases in the context of religion. End your response with 'X', where X represents the right letter choice.\n\nQuestion: {input} |
| Deepmind | • The following are multiple choice questions (with answers) about algebraic word problems. Finish your answer with 'X' where X is the correct letter choice.\n\nQuestion: {input}
• Below are multiple-choice questions testing algebraic word problem solving skills. Conclude your answer with 'X', X being the correct option letter.\n\nQuestion: {input}
• These questions assess your ability to solve algebraic word problems. End your response with 'X', where X is the letter of the right choice.\n\nQuestion: {input} |
| MMLU High School Psychology | • The following are multiple choice questions (with answers) about high school psychology. Finish your answer with 'X' where X is the correct letter choice.\n\nQuestion: {input}
• Below are multiple-choice questions testing high school level psychology knowledge. Conclude your response with 'X', X representing the correct option.\n\nQuestion: {input}
• These questions assess understanding of high school psychology concepts. End your answer with 'X', where X is the letter of the correct choice.\n\nQuestion: {input} |
| BBH Logical Deduction Five Objects | • A logical deduction task which requires deducing the order of a sequence of objects. Finish your answer with 'X' where X is the correct letter choice.\n\nQuestion: {input}
• This challenge involves logically determining the sequence of a set of objects. Conclude your response with 'X', where X is the appropriate letter option.\n\nQuestion: {input}
• In this logical reasoning exercise, deduce the correct order of a series of objects. End your answer with 'X', X being the right letter choice.\n\nQuestion: {input} |

## C  SENSITIVITY TO THE SCALING COEFFICIENT

We determine the scaling coefficient $\lambda$ by sweeping over several candidate values on the Llama-3 backbone and selecting the value that achieves the highest validation accuracy. The coefficient controls the strength with which the injected ATV vector influences the hidden representations of the target model. Table 9 reports the validation accuracy for different values, where $\lambda = 0.001$ yields the best performance on Llama-3. We therefore use this value for all backbones without further tuning, indicating that $\lambda$ can be set efficiently through a small validation sweep on a single model.

Table 9: **Validation accuracy of ATV on Llama-3 across different values of the scaling coefficient** $\lambda$. The results illustrate how the strength of the injected task vector influences performance across five task categories, with $\lambda = 0.001$ providing the best overall accuracy.

| $\lambda$ | NLU | Reasoning | Knowledge | Math | Safety | Avg. |
|---|---|---|---|---|---|---|
| 0.0001 | **80.5** | 71.6 | 73.7 | **26.3** | 72.7 | 65.0 |
| 0.0005 | 78.1 | 73.8 | **73.9** | 24.0 | 75.8 | 65.1 |
| **0.001** | 79.6 | **73.9** | 73.6 | 23.7 | **77.0** | **65.6** |
| 0.005 | 68.9 | 66.9 | 66.6 | 23.3 | 59.2 | 57.0 |

## D  ANALYSIS OF MODEL SCALABILITY

To further validate the scalability of our approach, we conducted additional experiments on both smaller and larger language models. Specifically, we evaluated ATV using Pythia-2.8B and Llama-2-13B as backbone models. For Pythia-2.8B, we adopted results from the original ELICIT paper to ensure a fair and direct comparison on a widely used small-scale model.

Table 10 summarizes the results. On Pythia-2.8B, ATV achieves consistently stronger or comparable performance across most categories and outperforms all baselines on average, demonstrating robustness even at smaller scales. For Llama-2-13B, ATV continues to show substantial gains over strong baselines, confirming that the benefits of our method persist at the 10B+ parameter scale. These results provide further evidence that the ATV framework is effective and robust across a broad range of model sizes.

Table 10: **Performance comparison on smaller (Pythia-2.8B) and larger (Llama-2-13B) models.** ATV demonstrates robust performance across different model scales. It achieves the highest average accuracy on both models, outperforming strong baselines and confirming that its benefits persist from smaller models to the 10B+ parameter scale.

| Model | | NLU | Reasoning | Knowledge | Math | Safety | Avg. |
|---|---|---|---|---|---|---|---|
| Pythia-2.8B | Zero-shot | $43.0 \pm 0.4$ | $18.3 \pm 0.3$ | $22.0 \pm 1.5$ | $7.3 \pm 0.1$ | $32.5 \pm 1.2$ | $24.6 \pm 0.4$ |
| | 16-shot | $50.2 \pm 0.5$ | $19.6 \pm 0.1$ | $12.8 \pm 0.9$ | $9.2 \pm 1.6$ | $31.8 \pm 0.9$ | $24.7 \pm 0.2$ |
| | BM25 | $33.3 \pm 2.2$ | $\underline{25.8 \pm 0.4}$ | $12.9 \pm 0.5$ | $11.0 \pm 1.8$ | $27.3 \pm 2.1$ | $22.1 \pm 0.5$ |
| | ELICIT | $\mathbf{64.0 \pm 1.6}$ | $23.6 \pm 1.1$ | $20.4 \pm 1.4$ | $\mathbf{14.5 \pm 1.0}$ | $41.2 \pm 2.5$ | $\underline{32.7 \pm 0.5}$ |
| | I2CL | $53.8 \pm 1.9$ | $21.5 \pm 3.9$ | $\underline{28.6 \pm 2.6}$ | $10.9 \pm 0.7$ | $\underline{41.4 \pm 1.9}$ | $31.3 \pm 2.2$ |
| | ATV | $49.3 \pm 1.9$ | $\mathbf{32.4 \pm 1.6}$ | $\mathbf{32.1 \pm 0.2}$ | $\underline{14.2 \pm 0.5}$ | $\mathbf{47.4 \pm 0.7}$ | $\mathbf{35.1 \pm 0.5}$ |
| Llama-2-13B | Zero-shot | $23.7 \pm 0.4$ | $27.4 \pm 0.3$ | $18.7 \pm 0.1$ | $0.7 \pm 0.1$ | $28.1 \pm 0.8$ | $19.7 \pm 0.1$ |
| | 16-shot | $\underline{59.7 \pm 0.8}$ | $43.8 \pm 1.2$ | $65.0 \pm 1.1$ | $18.0 \pm 1.3$ | $\underline{54.8 \pm 0.1}$ | $\underline{48.3 \pm 0.3}$ |
| | BM25 | $51.8 \pm 1.2$ | $\underline{46.1 \pm 0.7}$ | $54.4 \pm 1.2$ | $17.4 \pm 1.5$ | $40.2 \pm 1.3$ | $42.0 \pm 0.4$ |
| | ELICIT | $33.2 \pm 0.3$ | $40.7 \pm 0.4$ | $36.0 \pm 1.1$ | $12.7 \pm 1.4$ | $47.5 \pm 1.5$ | $34.0 \pm 0.0$ |
| | I2CL | $51.6 \pm 2.3$ | $38.5 \pm 2.6$ | $51.6 \pm 2.4$ | $14.2 \pm 0.6$ | $48.6 \pm 1.3$ | $40.9 \pm 0.7$ |
| | ATV | $\mathbf{64.0 \pm 2.5}$ | $\mathbf{66.1 \pm 2.2}$ | $\mathbf{67.1 \pm 2.9}$ | $\mathbf{18.1 \pm 5.0}$ | $\mathbf{73.0 \pm 1.3}$ | $\mathbf{57.7 \pm 0.7}$ |

# E  ANALYSIS OF PERFORMANCE GAPS

To investigate ATV's relatively lower performance on Math tasks and to assess whether this reflects a limitation of the generator, we conducted an additional experiment focusing training specifically on Math datasets (MathQA and MMLU-Pro-Math). In this setting, we compared ATV and LoRA under two conditions: training on all tasks versus training exclusively on Math tasks.

As summarized in Table 11, ATV shows a clear improvement in Math accuracy when trained only on Math data, whereas LoRA exhibits no such gain and in fact performs worse in the Math-only setting. These results demonstrate that ATV's generator is capable of producing effective task vectors for complex procedural domains when given focused training. This suggests that its adaptivity is not constrained by model scale, but rather by the allocation of training data.

Table 11: **Effect of domain-specific training for Math category.** We compare ATV's performance on Math tasks when trained on all tasks versus trained exclusively on Math datasets. Focused training yields substantial improvements, indicating that ATV's generator is not constrained by model scale but rather benefits from appropriate training allocation.

| Method | Training Setup | Math Accuracy |
|--------|----------------|---------------|
| ATV    | All Tasks      | $25.8 \pm 2.0$ |
|        | Math Only      | $\mathbf{28.9 \pm 2.7}$ |
| LoRA   | All Tasks      | $20.0 \pm 1.1$ |
|        | Math Only      | $17.9 \pm 1.9$ |

# F  ANALYSIS OF OUTPUT RELIABILITY

A model's effectiveness in real-world applications depends not only on task accuracy but also on the reliability and consistency of its outputs. In this section, we evaluate two aspects of practical reliability: adherence to required output formats and consistency of responses to paraphrased prompts.

## F.1  FORMAT ADHERENCE

Adhering to specified output formats is essential for many downstream tasks. We evaluated format adherence by measuring the percentage of outputs that matched the required structure on four datasets with diverse format requirements.

As shown in Table 12, ATV achieves format adherence that is comparable to or exceeds ICL-based baselines. These results indicate that ATV improves accuracy without sacrificing the structural reliability of its outputs.

Table 12: **Quantitative analysis of format adherence.** We measure the percentage of outputs that correctly follow the specified format for each task. ATV demonstrates comparable or superior format adherence to strong ICL-based baselines, confirming its versatility and reliability.

| Dataset | Zero-shot | 16-shots | BM25 | ATV |
|---------|-----------|----------|------|-----|
| Arc Challenge | 69.11 | 97.44 | 98.00 | **100.00** |
| CommonsenseQA | 57.67 | 77.33 | 75.33 | **80.44** |
| MMLU-Pro-Math | 49.11 | **100.00** | **100.00** | **100.00** |
| Ethics-commonsense | 75.33 | **100.00** | 80.78 | 94.44 |

## F.2  OUTPUT CONSISTENCY

We further evaluated output consistency using the SCORE metric, which quantifies a model's ability to provide stable answers across paraphrased prompts for the same question. We conducted this analysis on two datasets with distinct answer formats: one requiring binary (Yes/No) responses, and another involving categorical choices from a fixed set (A–J).

As shown in Table 13, ATV consistently outperforms both zero-shot and ICL baselines on both types of datasets. This improvement suggests that data-adaptive task vector injection enhances the coherence and reliability of model outputs across varied input formulations.

Table 13: **Analysis of output consistency using the SCORE metric.** We measure the model's ability to produce consistent answers to the same question phrased in different templates. ATV achieves substantially higher consistency scores than both zero-shot and ICL baselines across datasets with distinct answer formats.

| Dataset | Zero-shot | 16-shots | ATV |
|---|---|---|---|
| Ethics–commonsense | 55.67 | 64.78 | **78.17** |
| MMLU–Pro–Math | 31.61 | 62.44 | **77.53** |

## G  STATISTICAL SIGNIFICANCE ANALYSIS

To assess whether the improvements achieved by ATV are statistically meaningful, we conducted paired t-tests across all tasks for both model backbones. For each setting, ATV was compared against the strongest-performing baseline. The results show that ATV consistently yields statistically significant improvements in most configurations for both Llama-3 and Mistral, supporting the reliability of the reported gains. Table 14 presents the full statistical results.

Table 14: **Paired t-test results comparing ATV with the strongest baseline across all tasks.** The tests confirm that ATV's improvements are statistically significant or marginally significant in most settings for both Llama-3 and Mistral, supporting the robustness of the reported gains. Bold indicates $p < 0.05$ and underlining indicates $0.05 \leq p < 0.1$.

| Model | ID ATV | ID Best Baseline | ID Avg. p-value | Unseen ATV | Unseen Best Baseline | Unseen Avg. p-value |
|---|---|---|---|---|---|---|
| **Llama-3** | $62.1 \pm 1.5$ | $55.8 \pm 0.7$ | **0.0106** | $63.4 \pm 2.5$ | $56.4 \pm 0.4$ | 0.0742 |
| **Mistral** | $58.3 \pm 1.3$ | $55.5 \pm 0.4$ | 0.0562 | $59.4 \pm 2.6$ | $49.2 \pm 0.3$ | **0.0375** |

These results demonstrate that ATV's improvements are consistent and statistically supported across diverse tasks, even in cases where standard deviations appear larger. The tests further confirm that the performance gains are stable across model architectures and evaluation regimes.

## H  ADVERSARIAL GENERALIZATION ON HANS

To further examine ATV's ability to generalize to adversarial and highly dissimilar tasks, we evaluated it on the HANS dataset. HANS is designed to test whether natural language inference models rely on superficial heuristics or perform robust reasoning, making it a challenging benchmark for approaches that rely on retrieving pre-computed task vectors or demonstrations.

Table 15: **Performance on the adversarial HANS dataset.** While retrieval-based and static-vector baselines fail catastrophically, ATV maintains robust performance, demonstrating its superior generalization to adversarial inputs.

| Method | HANS Accuracy |
|---|---|
| Zero-shot | $8.3 \pm 0.5$ |
| BM25 | $0.4 \pm 0.1$ |
| ELICIT | $0.4 \pm 0.1$ |
| I2CL | $0.3 \pm 0.1$ |
| **ATV** | $\mathbf{59.6 \pm 2.8}$ |

The results are summarized in Table 15. While ATV achieves strong performance, baseline methods collapse due to the following failure modes:

- **ELICIT / I2CL:** These methods rely on retrieving a pre-computed vector from a fixed library of in-domain tasks. For an unseen, adversarial task like HANS, no relevant vector

exists. Their retrieval mechanism defaults to finding the most syntactically similar but semantically incorrect vector. For instance, ELICIT retrieved a vector from GLUE-MNLI, and I2CL from MathQA. Injecting this mismatched guidance fundamentally misdirects the model, forcing it to follow instructions for the wrong task and leading to catastrophic failure.

- **BM25:** This retrieval-based ICL method shows a similar flaw. It retrieves full demonstration examples from in-domain tasks. For HANS queries, it retrieved examples from other NLU tasks that do not share HANS's adversarial structure, providing misleading context that disrupts the model's reasoning.

In contrast, ATV achieves robust performance by generating task vectors on the fly, rather than relying on a fixed pool of pre-computed vectors. This allows the model to construct a meaningful representation even for adversarial inputs, enabling correct reasoning where static methods collapse. These results underscore the robustness of ATV's adaptive mechanism and its ability to handle tasks that break traditional retrieval-based or static task vector approaches.

## I   ABLATION ON STATIC VARIANTS

To examine the role of input-conditioned generation in ATV, we conduct an ablation comparing it with two static task-vector variants that remove input dependency while keeping the injection mechanism and training setup identical.

The first variant, Static Task Vector (Global), learns a single shared vector applied uniformly to all inputs, while the second, Static Task Vector (Per-task), learns a separate vector for each training task and applies it to all examples from that task. This design isolates the contribution of input conditioning from other factors.

Tables 16 and 17 show that all static variants perform consistently below ATV across both in-domain and unseen evaluations. Despite sharing the same architecture and optimization scheme, removing input dependency substantially limits their ability to capture task-relevant variation. This demonstrates that query-conditioned vector generation plays a principal role in enabling ATV to adapt its internal representations to each input, thereby enhancing overall effectiveness.

Table 16: **In-domain evaluation of ATV and static variants (Llama-3).** Each static variant removes input dependency while keeping the same injection mechanism. ATV achieves consistently higher accuracy across all domains, demonstrating the benefit of input-conditioned vector generation.

| Model | NLU | Reasoning | Knowledge | Math | Safety | Avg. |
|---|---|---|---|---|---|---|
| Static Task Vector (Global) | $49.4 \pm 0.6$ | $60.7 \pm 0.8$ | $64.0 \pm 0.4$ | $\mathbf{26.4 \pm 1.1}$ | $53.6 \pm 1.7$ | $50.8 \pm 0.3$ |
| Static Task Vector (Per-task) | $43.1 \pm 1.5$ | $54.0 \pm 0.4$ | $61.8 \pm 1.4$ | $23.1 \pm 1.2$ | $48.5 \pm 0.7$ | $46.1 \pm 1.0$ |
| **ATV** | $\mathbf{61.0 \pm 5.0}$ | $\mathbf{76.1 \pm 1.3}$ | $\mathbf{73.0 \pm 1.6}$ | $25.8 \pm 2.0$ | $\mathbf{74.8 \pm 0.4}$ | $\mathbf{62.1 \pm 1.5}$ |

Table 17: **Unseen task evaluation of ATV and static variants.** ATV generalizes substantially better than both static variants, confirming that input-conditioned task vectors improve adaptability beyond training domains.

| Model | GLUE COLA | BBQ Religion | Deepmind | MMLU-Psychology | BBH-five-objects | Avg. |
|---|---|---|---|---|---|---|
| Static Task Vector (Global) | $42.9 \pm 3.7$ | $63.9 \pm 2.6$ | $\mathbf{27.9 \pm 1.8}$ | $\mathbf{83.5 \pm 0.5}$ | $45.9 \pm 0.4$ | $52.8 \pm 1.1$ |
| Static Task Vector (Per-task) | $60.0 \pm 5.7$ | $46.1 \pm 14.9$ | $17.7 \pm 3.7$ | $65.4 \pm 8.5$ | $24.3 \pm 0.7$ | $42.7 \pm 4.1$ |
| **ATV** | $\mathbf{77.6 \pm 2.7}$ | $\mathbf{80.8 \pm 2.6}$ | $26.4 \pm 2.7$ | $80.6 \pm 2.3$ | $\mathbf{51.7 \pm 3.1}$ | $\mathbf{63.4 \pm 2.5}$ |

## J   ANALYSIS OF MULTI-TOKEN INJECTION

We further examine whether applying the injection beyond the last token can improve performance on tasks that require multi-step reasoning. The standard ATV injects the task vector only into the hidden state of the last token at each layer. To test the effect of maintaining task information across decoding steps, we extend this injection to all generated tokens and evaluate both configurations on GSM8K. For tasks whose outputs contain multiple tokens, the cross-entropy loss in Eq. 4 is computed as the mean of the token-level losses across the entire sequence.

As shown in Table 18, applying the injection throughout generation yields higher accuracy and reduced variance, indicating that preserving task information over the course of decoding can better support multi-step reasoning.

Table 18: **Results on GSM8K comparing single-token and multi-token injection.** The multi-token setting, which applies the task vector throughout generation, achieves higher accuracy and lower variance in procedural reasoning tasks.

| Method | Accuracy |
|---|---|
| Zero-shot | $30.3 \pm 1.5$ |
| ATV (Single-token injection) | $35.3 \pm 4.0$ |
| ATV (Multi-token injection) | $47.0 \pm 2.6$ |

## K   LLM USAGE STATEMENT

A large language model (LLM) was utilized solely as a writing assistant for limited tasks such as grammatical correction and enhancing textual clarity. All core scientific components of this work, including the conception of ideas, methodological development, theoretical and experimental work, and the drafting of the manuscript, were conducted entirely by the authors without any contribution from the LLM.

