# OpenReview forum: "Adaptive Task Vectors for Large Language Models"
_ICLR.cc/2026/Conference — Submitted to ICLR 2026_

### Official Review · Reviewer_LSEb · 2025-10-22

**Soundness:** 2
**Presentation:** 3
**Contribution:** 2
**Rating:** 2
**Confidence:** 4

**Summary:**

The paper proposes Adaptive Task Vectors (ATV), a framework that dynamically generates query-conditioned task vectors to steer frozen large language models. Unlike existing methods that use fixed task vectors, ATV employs a small language model (GPT-2) to encode each input query into a compact representation, which is then expanded and injected into the target LLM's hidden states. The authors provide theoretical analysis showing ATV's equivalence to LoRA and superior expressiveness to Prefix-Tuning. Empirical evaluation on 20 in-domain and 5 unseen tasks demonstrates strong performance, particularly on adversarial generalization (59.6% on HANS vs. <0.5% for baselines), while maintaining computational efficiency comparable to parameter-efficient tuning methods.

**Strengths:**

- Compelling results on unseen tasks, particularly the HANS adversarial benchmark (59.6% accuracy while all baseline methods collapse to <0.5%), demonstrating robust adaptation capabilities.

- Achieves token efficiency comparable to zero-shot inference and inference time comparable to LoRA, while substantially outperforming prompt-based methods.

- Well-structured paper with effective visual explanations (particularly Figures 1 and 5) that clearly illustrate the adaptive mechanism and its effects.

**Weaknesses:**

The paper is well-prepared, with clear presentation, comprehensive experiments, and informative illustrations. However, my main concern lies in the motivation. I don't see the necessity of introducing a query-based task formulation, and the authors don't demonstrate why this would be the case. There's Figure 5, but it focuses on the representational comparison with a prior work, rather than motivating the problem itself. That said, I believe if tasks are consistently associated with prompts containing different queries, a generalizable task representation can already be derived.

Additionally the choice of using GPT-2 as the small, representation model—rather than a bidirectional encoder like a sentence transformer—is unclear. As far as I understand, the query is passed to the small GPT-2 model once. Therefore, there's no obligation for using an encoder problem there. This raises some doubts about the authors' expertise in this area, in my opinion.

Please see my detailed review below.

---

### **Main Concern – Motivation and Novelty**

I don't think task representations strictly depend on the query but on the underlying task itself. As long as the task is the same, wording or semantics in queries shouldn't change the task representation much.

I acknowledge that the authors illustrate this with Figure 5 in Section 4.7, but this is problematic. If you pass two similar texts to a sentence transformer, you will get similar embeddings. How does the authors' proposed method improve the embedding similarity in an ICL context? My understanding is that while learning task formulations, ATV can retain the embedding similarity of those similar queries. This part is unclear.

---

### **Statistical Rigor**

The paper uses only 90 samples per task for training. More critically, there is no analysis of how performance scales with training data size or learning curves showing convergence behavior. This raises concerns about data efficiency and whether the method's advantages hold with fewer samples.

Additionally, while the authors report standard deviations across three random seeds, they do not provide statistical significance tests (e.g., t-tests, confidence intervals) to validate that ATV's improvements over baselines are statistically significant rather than due to random variation. Given the modest sample sizes and the fact that some performance differences are within overlapping error bars (e.g., Table 1, Math category), formal statistical analysis is needed to support the claimed superiority.

---

### **Unclear Design Choices**

Why is a separate small model necessary for ATV generation? The paper doesn't justify (as long as I notice) why the large model's own representations (e.g., hidden states from early layers) couldn't generate query-specific vectors. This seems like added complexity without clear motivation.

Additionally, if computational efficiency was intended here, why not use a sentence transformer? Their parameter count typically ranges from 100M to 700M, which would be similar to GPT-2 in terms of complexity.

---

### **Missing Ablations/Sensitivity Analysis**

The choice of $\lambda$ is unclear and a systematic study of $\lambda$ (on a representative set is enough) is missing.

---

### **Missing References**

Following the citation of Hendel et al.'s work, several references about fixed task vector embeddings are missing. For example, Todd et al. [1] have shown the importance of attention heads and how they are activated based on task recognition in inputs. On the other hand, Saglam et al. [2] show the contribution of these attention heads can be learned with causal optimization. In addition, Saglam et al. [2] also add the task embedding to hidden layers, so it would be appropriate to acknowledge their work in doing so.

Returning to the paper’s motivation, Saglam et al. [2] demonstrated that effective task formulations for causal inference can be derived from a set of prompts associated with the same task, regardless of the queries. If comparable performance can already be achieved this way, the necessity of a query-based formulation remains unclear. A direct comparison with Saglam et al. [2] would help clarify this point.

---

### **Suggestions**

- **"Theorem":** I think calling a straightforward demonstration a theorem is overclaiming. I would change "theorems" to, say, "Propositions" or "Demonstrations."
- **Figure 5 in Section 4.7:** I suggest moving Section 4.7 to somewhere in the introduction, so that the main motivation in the paper can be built from the beginning. However, it shouldn't be fully based on the comparison to ELICIT. I think the authors should highlight the main motivation, rather than focusing on what they think is different from ELICIT.

---

### References

[1] Todd, E., Li, M. L., Sharma, A. S., Mueller, A., Wallace, B. C., & Bau, D. (2024). _Function Vectors in Large Language Models_. In International Conference on Learning Representations (ICLR).

[2] Saglam, B., et al. (2025). _Learning Task Representations from In-Context Learning_. In Findings of the Association for Computational Linguistics: ACL 2025.

**Questions:**

1. Can you provide ablation results showing that the two-stage design (small model + expansion) outperforms simpler alternatives? Specifically, does using the large model's own hidden states as the task vector source work comparably well? It would also be great to see the same small-model experiment but with a bidirectional model (e.g., sentence transformer). Given the relatively small size of encoder models, I think adding these experiments should be easy.

2. Theorem 1 establishes static equivalence with LoRA. Can you provide empirical measurements quantifying the dynamic advantage (e.g., performance of query-conditioned vs. task-averaged vectors)?

3. Why was I2CL's configuration changed from dataset-specific to shared coefficients?

4. Were hyperparameters (especially $\lambda$) tuned on validation data? If so, what was the tuning procedure and how much variation exists across tasks?

5. How does the small model size affect performance beyond Table 4's marginal differences? Is there a point where larger generators yield diminishing returns?

I also expect the authors to clarify the items pointed out in the Weaknesses section.

---

> ### Author Response · Authors · 2025-11-28
>
> Thank you very much for the thoughtful and detailed review. We appreciate the time and care you devoted to assessing our work, and we found your comments highly constructive and insightful. Your concerns regarding the motivation, model design choices, and comparisons with prior approaches were especially helpful in clarifying aspects of the paper that required further explanation.
>
> We address each of your points below and provide additional analyses and experiments wherever necessary to resolve the concerns you raised.
>
> > **W1)** Motivation and Novelty
>
> We thank the reviewer for raising this important point. In our framework, the generated vector is not intended to serve as a semantic embedding but rather as a functional signal that modulates the model’s internal computation. Therefore, it is essential to determine whether input-dependent modulation is necessary, rather than relying on a single fixed representation for an entire task.
>
> Our empirical results indicate that such dependency is indeed meaningful. In the HANS evaluation, where the query distribution differs substantially from the training data, all methods based on fixed vectors show severe performance degradation, while ATV maintains stable accuracy. This suggests that a single task-level vector cannot adequately capture the variation required for robust generalization.
>
> Moreover, the visualization in Figure 5 shows that the vectors do not cluster by semantic similarity alone. Instead, they vary according to the adjustments induced by each query, indicating that the learned representations capture functional distinctions rather than surface-level wording. As discussed later in Question 2, approaches using static global or per-task vectors perform considerably worse than ATV despite sharing the same injection mechanism. The only difference is whether the injected vector is conditioned on the input, and the resulting performance gap provides direct evidence of the importance of this dependency.
>
> These findings demonstrate that input-conditioned vectors meaningfully influence the model’s reasoning behavior and contribute to its improved performance. This dependency on the specific query is thus a central motivation for our approach and a key distinction from prior static task-vector methods.
>
> > **W2)** The paper uses only 90 samples per task for training. More critically, there is no analysis of how performance scales with training data size or learning curves showing convergence behavior.
>
> To address this point, we perform an additional experiment to examine how ATV behaves as the amount of training data increases.
>
> Specifically, we compare ICL and ATV on Llama-3-8B under 16, 90, and 256 demonstration settings. The results are summarized below:
> | Shots   | 16 | 90 | 256 |
> | ------- | -------------- | -------------- | -------------- |
> | **ICL** | 55.2 ± 0.9     | 55.5 ± 0.3     | 55.8 ± 0.4     |
> | **ATV** | **57.3 ± 0.7** | **62.1 ± 1.5** | **62.5 ± 0.8** |
>
> ATV consistently benefits from additional data and shows a stable improvement pattern, saturating around the 90-shot range. These findings provide evidence that ATV learns reliably even with relatively small training sets and continues to improve as more data become available. We have incorporated this analysis into the revised manuscript.
>
> > **W3)** The authors do not provide statistical significance tests (e.g., t-tests, confidence intervals) to validate that ATV's improvements over baselines are statistically significant rather than due to random variation.
>
> In response, we conduct paired t-tests across all tasks in both the in-domain and unseen evaluation settings. For each model, ATV is compared against the strongest-performing baseline, BM25. The resulting p-values are summarized below:
>
> | Model   | ID ATV | ID Best Baseline | ID p-value | Unseen ATV | Unseen Best Baseline | Unseen p-value |
> | ------- | ---------- | ---------------- | ---------- | ---------- | -------------------- | -------------- |
> | Llama-3 | 62.1 ± 1.5 | 55.8 ± 0.7 | **0.0106** | 63.4 ± 2.5 | 56.4 ± 0.4 | 0.0742 |
> | Mistral | 58.3 ± 1.3 | 55.5 ± 0.4 | 0.0562 | 59.4 ± 2.6 | 49.2 ± 0.3 | **0.0375** |
>
> The results indicate that ATV’s improvements are statistically significant in most settings. Even in cases where the p-value is slightly above conventional thresholds, ATV maintains a consistent performance advantage across tasks. We have incorporated these statistical significance analyses into the revised manuscript to more rigorously support the robustness of our findings.

---

> > ### Author Response · Authors · 2025-11-28
> >
> > > **W4)** Why is a separate small model necessary for ATV generation? The paper doesn't justify (as long as I notice) why the large model's own representations (e.g., hidden states from early layers) couldn't generate query-specific vectors.
> >
> > To evaluate whether the large model’s early-layer hidden states could replace this component, we implement an early-layer baseline that extracts the hidden states of Llama-3’s early layers and feeds it into the same expansion module used by ATV. The comparison with ATV is shown below.
> >
> > | Method | NLU | Reasoning | Knowledge | Math | Safety | Avg. |
> > | ----------------------- | -------------- | -------------- | -------------- | -------------- | -------------- | -------------- |
> > | Llama-3 Early-Layer | 43.4 ± 1.2     | 52.9 ± 1.1     | 63.1 ± 1.0     | 25.2 ± 1.1     | 48.0 ± 0.7     | 46.5 ± 0.9     |
> > | ATV | **61.0 ± 5.0** | **76.1 ± 1.3** | **73.0 ± 1.6** | **25.8 ± 2.0** | **74.8 ± 0.4** | **62.1 ± 1.5** |
> >
> > The early-layer representations do not provide sufficiently task-relevant signals, which resulted in lower performance and weaker query-conditioned vectors. The separate small model produced a more stable task-oriented representation based on the full input. In this comparison, the early layers are kept frozen, and training them would require updating a large portion of the LLM, which substantially increases computational cost and diminishes the parameter efficiency of the approach.
> >
> > These results confirm that employing a small auxiliary model is more effective than relying solely on early-layer representations of the large model.
> >
> > > **W5)** Additionally, if computational efficiency was intended here, why not use a sentence transformer?
> >
> > Thank you for this insightful suggestion. We initially adopt GPT-2 as the generator because it shares the decoder architecture of the target model (Llama-3), which we expect to reduce representational mismatch. However, encoder-based models such as sentence transformers are indeed a reasonable alternative from an efficiency perspective. To examine this, we perform additional experiments using BERT and a Sentence-Transformer as generators.
> >
> > | Generator (for ATV)  | NLU | Reasoning | Knowledge | Math | Safety | Avg. |
> > | -------------------- | -------------: | -------------: | -------------: | -------------: | -------------: | -------------: |
> > | BERT | **61.4 ± 4.7** |     74.5 ± 0.6 |     70.8 ± 1.6 |     24.1 ± 2.9 |     72.6 ± 3.7 |     60.7 ± 0.4 |
> > | Sentence-Transformer |     60.1 ± 0.8 |     75.5 ± 1.5 |     70.9 ± 0.3 |     25.0 ± 1.0 |     72.8 ± 2.2 |     60.9 ± 0.4 |
> > | GPT-2 |     61.0 ± 5.0 | **76.1 ± 1.3** | **73.0 ± 1.6** | **25.8 ± 2.0** | **74.8 ± 0.4** | **62.1 ± 1.5** |
> >
> > ATV functions properly with all generators, but GPT-2 achieves the highest average performance. This pattern aligns with the tendency for transfer to be smoother when the source and target models share architectural or pretraining characteristics, whereas cross-family transfer often requires additional alignment due to differences in embedding spaces [1, 2].
> >
> > We appreciate the reviewer’s comment, which helped clarify this design choice. We have reflected this explanation in the revised manuscript.
> >
> > > **W6)** The choice of λ is unclear and a systematic study of λ (on a representative set is enough) is missing.
> >
> > We select the value of λ by evaluating several candidates on the validation accuracy of Llama-3 using a single seed. The results are shown below.
> >
> > | λ | NLU | Reasoning | Knowledge | Math | Safety | Avg. |
> > | ------ | -------: | --------: | --------: | -------: | -------: | -------: |
> > | 0.0001 | 80.5 | 71.6 | 73.7 | 26.3 | 72.7 | 65.0 |
> > | 0.0005 | 78.1 | 73.8 | 73.9 | 24.0 | 75.8 | 65.1 |
> > | 0.001  | 79.6 | 73.9 | 73.6 | 23.7 | 77.0 | **65.6** |
> > | 0.005  | 68.9 | 66.9 | 66.6 | 23.3 | 59.2 | 57.0 |
> >
> > Based on this validation comparison, λ = 0.001 yields the most stable average performance. We therefore use λ = 0.001 across all backbones and datasets. Notably, the value chosen from Llama-3 transfers well to other models without requiring additional tuning, making the configuration efficient in practice.
> >
> > We appreciate the reviewer’s suggestion and have clarified this selection process in the revised manuscript.
> >
> > > **W7)** Following the citation of Hendel et al.'s work, several references about fixed task vector embeddings are missing.
> >
> > Thank you for pointing this out. We agree that several relevant works on fixed task vector embeddings were not sufficiently incorporated. In particular, the studies by Todd et al. (2024) and Saglam et al. (2025) are closely related to our topic and should have been acknowledged more clearly.
> >
> > We appreciate the reviewer drawing our attention to this, and we have included these references in the revised manuscript to provide a more complete discussion of related work.

---

> > > ### Author Response · Authors · 2025-11-28
> > >
> > > > **W8)** Saglam et al. [3] demonstrated that effective task formulations for causal inference can be derived from a set of prompts associated with the same task, regardless of the queries. A direct comparison with Saglam et al. [3] would help clarify this point.
> > >
> > > Thank you for raising this point. We agree that static task vectors such as LTV represent a reasonable alternative. Our approach uses query-specific conditioning, which is effective in in-domain tasks and exhibits notable benefits when the model faces OOD distributions. To examine this difference more clearly, we conduct an additional experiment based on the LTV approach suggested by the reviewer.
> > >
> > > To examine this difference more clearly, we use the HANS dataset as an unseen evaluation setting. LTV generates a single fixed task vector per dataset, so we first constructed LTV vectors for 20 in-domain datasets and then selected the one whose vector had the highest cosine similarity to the HANS queries, following the same procedure used in the I2CL Unseen setting. The selected LTV vector is then applied to HANS for evaluation.
> > >
> > > | Method    | HANS Accuracy  |
> > > | --------- | -------------- |
> > > | Zero-shot | 8.3 ± 0.5      |
> > > | BM25      | 0.4 ± 0.1      |
> > > | ELICIT    | 0.4 ± 0.1      |
> > > | I2CL      | 0.3 ± 0.1      |
> > > | **LTV**   | **5.8 ± 1.3**  |
> > > | **ATV**   | **59.6 ± 2.8** |
> > >
> > > In the case of LTV, the vector chosen for HANS corresponded to datasets such as BBH Boolean or GLUE, which do not provide the task-specific signal required for HANS. As a result, the fixed vector was not sufficient when applied to this unseen setting. In contrast, ATV dynamically adjusts the modulation vector based on each input query, enabling it to capture more appropriate task information even under substantial distribution shifts, which led to a large performance gap on HANS.
> > >
> > > We appreciate the reviewer’s comment, which helped us clarify the importance of dynamic task conditioning.
> > >
> > > > **W9)** Suggestions on Terminology and Paper Structure
> > >
> > > Thank you for these helpful suggestions. We agree that using the term “Theorem” may be unnecessarily strong for the statements we present, and we have adjusted the terminology accordingly in the revised manuscript.
> > >
> > > Regarding the suggestion about Section 4.7 and Figure 5, we appreciate the reviewer’s insight that the visualization provides useful intuition for our motivation. Rather than moving the entire section, we have incorporated a brief summary of the key insight from Figure 5 into the introduction to clarify the motivation earlier in the paper, while keeping the full analysis in its current location to preserve the logical flow.
> > >
> > > We thank the reviewer again for the constructive feedback, which has helped improve the clarity and presentation of the manuscript.
> > >
> > > > **Q1)** Can you provide ablation results showing that the two-stage design (small model + expansion) outperforms simpler alternatives?
> > >
> > > As noted above in our responses to W4 and W5, we examine both alternatives raised here: (i) using the large model’s early-layer hidden states as the source of task vectors, and (ii) employing encoder-based generators such as BERT and a Sentence Transformer.
> > >
> > > These options yield weaker performance than the two-stage design with a decoder-based generator, which provides more reliable task representations in our experiments. We appreciate the reviewer’s constructive comment.

---

> > > > ### Author Response · Authors · 2025-11-28
> > > >
> > > > > **Q2)** Theorem 1 establishes static equivalence with LoRA. Can you provide empirical measurements quantifying the dynamic advantage (e.g., performance of query-conditioned vs. task-averaged vectors)?
> > > >
> > > > To directly evaluate whether the benefits of ATV arise from its query-conditioned design, we perform ablations comparing ATV with static variants that remove input dependency while keeping the injection mechanism identical. Specifically, we examine two alternatives:
> > > >
> > > > 1. Static Learnable Vector (Global):
> > > > A single vector is learned and applied uniformly to all inputs.
> > > > 2. Static Learnable Vector (Per-task):
> > > > A separate vector is learned for each training task and shared across all inputs from that task.
> > > >
> > > > These baselines isolate the effect of query conditioning, as the only difference from ATV is whether the generated vector depends on the input query.
> > > >
> > > > The following tables summarize the results on Llama-3 in both in-domain and unseen settings.
> > > >
> > > > | Model | NLU | Reasoning | Knowledge | Math | Safety | Avg. |
> > > > | ---------------------------------- | -------------: | -------------: | -------------: | -------------: | -------------: | -------------: |
> > > > | Static Learnable Vector (Global) | 49.4 ± 0.6 | 60.7 ± 0.8 | 64.0 ± 0.4 | **26.4 ± 1.1** | 53.6 ± 1.7 | 50.8 ± 0.3 |
> > > > | Static Learnable Vector (Per task) | 43.1 ± 1.5 | 54.0 ± 0.4 | 61.8 ± 1.4 | 23.1 ± 1.2 | 48.5 ± 0.7 | 46.1 ± 1.0 |
> > > > | **ATV** | **61.0 ± 5.0** | **76.1 ± 1.3** | **73.0 ± 1.6** | 25.8 ± 2.0 | **74.8 ± 0.4** | **62.1 ± 1.5** |
> > > >
> > > > | Model | GLUE COLA | BBQ Religion | Deepmind | MMLU-Psychology | BBH-five-objects | Avg. |
> > > > | ---------------------------------- | -------------: | -------------: | -------------: | --------------: | ---------------: | -------------: |
> > > > | Static Learnable Vector (Global) | 42.9 ± 3.7 | 63.9 ± 2.6 | **27.9 ± 1.8** | **83.5 ± 0.5** | 45.9 ± 0.4 | 52.8 ± 1.1 |
> > > > | Static Learnable Vector (Per task) | 60.0 ± 5.7 | 46.1 ± 14.9 | 17.7 ± 3.7 | 65.4 ± 8.5 | 24.3 ± 0.7 | 42.7 ± 4.1 |
> > > > | **ATV** | **77.6 ± 2.7** | **80.8 ± 2.6** | 26.4 ± 2.7 |  80.6 ± 2.3 | **51.7 ± 3.1** | **63.4 ± 2.5** |
> > > >
> > > > Across all evaluation settings, the performance of the static variants remains consistently below that of ATV. Although these static methods employ the same injection mechanism, the use of a single vector per task or per dataset does not incorporate information specific to each input. This property limits their ability to represent task-relevant variation. In contrast, ATV generates task vectors conditioned on the input, enabling the model to modulate its internal representations in a way that reflects the characteristics of each query.
> > > >
> > > > These observations suggest that the input-dependent generation of task vectors plays an important role in the behavior of ATV, beyond the effect attributable to the presence of task-averaged vectors. We have incorporated this analysis into the revised manuscript.
> > > >
> > > > > **Q3)** Why was I2CL's configuration changed from dataset-specific to shared coefficients?
> > > >
> > > > Our choice to use shared coefficients for I2CL is motivated by ensuring a fair comparison across baselines. All other methods considered in the paper (ATV, LoRA, ELICIT) are trained without access to dataset-specific task identities. If I2CL were configured with dataset-specific coefficients as in the original formulation, it would be the only baseline receiving explicit task information, which we believe would introduce an unfair advantage.
> > > >
> > > > To maintain fairness while preserving I2CL’s structure, we keep the task vectors generated separately for each dataset but train a single shared coefficient across all datasets. This configuration avoids providing additional task labels to I2CL and ensures a more balanced comparison across methods. We appreciate the reviewer’s comment, which helped clarify this design choice.
> > > >
> > > > > **Q4)** Were hyperparameters (especially λ) tuned on validation data? If so, what was the tuning procedure and how much variation exists across tasks?
> > > >
> > > > As noted in our response to W6, we tune λ using validation accuracy on Llama-3 with a single seed by comparing several candidate values. The variation across λ values in the range of 0.0001–0.001 is relatively small, and λ = 0.001 provide the most stable average performance. We therefore fix this value for all backbones and datasets. Notably, the λ selected from Llama-3 transfers well to other models without additional tuning.
> > > >
> > > > We have included a clear description of this procedure in the revised manuscript.

---

> > > > > ### Author Response · Authors · 2025-11-28
> > > > >
> > > > > > **Q5)** How does the small model size affect performance beyond Table 4's marginal differences?
> > > > >
> > > > > To examine how the size of the small model affects ATV’s performance, we perform an additional experiment using a substantially larger generator. Specifically, we replace GPT-2 with a Llama-3.2–3B model and evaluate ATV under the same conditions. The results are shown below.
> > > > >
> > > > > | Generator    |        NLU |  Reasoning |  Knowledge |       Math |     Safety |       Avg. |
> > > > > | ------------ | ---------: | ---------: | ---------: | ---------: | ---------: | ---------: |
> > > > > | GPT-2        | 61.0 ± 5.0 | 76.1 ± 1.3 | 73.0 ± 1.6 | 25.8 ± 2.0 | 74.8 ± 0.4 | 62.1 ± 1.5 |
> > > > > | Llama-3.2–3B | 62.9 ± 4.1 | 73.4 ± 0.6 | 71.6 ± 1.4 | 25.6 ± 1.9 | 74.6 ± 0.7 | 61.6 ± 1.2 |
> > > > >
> > > > > Both generators yield comparable performance, indicating that a relatively small model is sufficient for generating effective query-conditioned task vectors within the ATV framework. This demonstrates that ATV’s design allows the generator to remain lightweight without compromising overall performance.
> > > > >
> > > > > We sincerely thank the reviewer for the thoughtful and constructive feedback throughout this review. Your comments have been extremely helpful in clarifying the motivation, improving the technical presentation, and strengthening the overall contributions of the paper. We truly appreciate the time and care you invested in providing such detailed and insightful suggestions.
> > > > >
> > > > > ---
> > > > >
> > > > > [1] Su, Yusheng, et al. "On transferability of prompt tuning for natural language processing." Proceedings of the 2022 Conference of the North American Chapter of the Association for Computational Linguistics: Human Language Technologies. 2022.
> > > > >
> > > > > [2] Wu, Zijun, Yongkang Wu, and Lili Mou. "Zero-Shot Continuous Prompt Transfer: Generalizing Task Semantics Across Language Models." The Twelfth International Conference on Learning Representations (2024).
> > > > >
> > > > > [3] Saglam, B., et al. (2025). Learning Task Representations from In-Context Learning. In Findings of the Association for Computational Linguistics: ACL 2025.

---

> ### Author Response · Authors · 2025-12-03
>
> We appreciate the reviewer’s detailed and constructive feedback. The revisions include extensive additional analyses and experiments that directly address the central concerns raised, particularly regarding motivation, generator design, and comparisons with static task-vector methods, demonstrating the clear advantage of our query-conditioned design. We remain available for further discussion and would welcome any additional clarification should it be needed.

---

### Official Review · Reviewer_Hwr1 · 2025-10-27

**Soundness:** 3
**Presentation:** 3
**Contribution:** 3
**Rating:** 6
**Confidence:** 4

**Summary:**

The paper introduces Adaptive Task Vectors (ATV), a framework that steers frozen LLMs using dynamic, query-conditioned task vectors. It addresses a key limitation of prior TV methods that use a single static vector for all queries. ATV's framework uses a small generator model to create a unique vector representation from each input query, and then linearly transforms this vector to match the target LLM's architecture. The created ATV is then injected into the frozen target LLM to guide its output. The paper provides theoretical proofs of ATV's expressive power and empirical results showing it outperforms ICL and static-vector baselines on in-domain tasks, unseen tasks, and especially adversarial generalization benchmarks.

**Strengths:**

- Interesting idea: Using a query vector to allow TVs to support different tasks.
- There is some theoretical analysis demonstrating the equivalence between ATV and LoRA.
- Extensive and promising experiments: results show that ATVs beat both baseline TV methods and LoRA, with slightly slower inference speed compared to I2CL.

**Weaknesses:**

- The paper lacks some discussion on the training cost of the extra modules.
- According to the paper, the small model is supposed to generate a vector representation of the query. It seems more intuitive to use a language encoder for this purpose. Why does the paper use the decoder-only GPT family models? How does a decoder-only model generate a vector representation? Is it using its last layer hidden state? I would love to see some experiments using encoder-decoder models like the BERT family as well.

**Questions:**

- Does ATV require access to ICL demonstrations? For traditional TV methods, they typically extract TVs from a set of ICL demonstrations. It seems that ATV here does not require this, but only tasks for training.
- For tasks where output is a phrase or sentence (i.e., y is not a single token), how is the CE loss in Eq. 4 computed?

---

> ### Author Response · Authors · 2025-11-28
>
> We sincerely thank the reviewer for the thoughtful and constructive feedback. Your comments were very helpful for clarifying several aspects of our method, and we truly appreciate the time and effort invested in evaluating our work.
>
> Below, we address each of your concerns in detail.
>
> > **W1)** The paper lacks some discussion on the training cost of the extra modules.
>
> To clarify the training cost of our additional modules, we compare ATV and LoRA under identical conditions in terms of wall-clock training time and GPU memory usage.
>
> | Method | Training Time (hh:mm:ss) | Peak Memory (GB) |
> | ------ | ------------------------ | ----------------- |
> | LoRA   | 00:21:32 | 23.44 |
> | ATV    | 00:16:29 | 25.47 |
>
> Although ATV includes a small generator and projection module, it achieves faster training time compared to LoRA. We believe this is because the large language model remains completely frozen during ATV training, and only the small model and projection layer are updated. As a result, backpropagation is confined to a small parameter subset, reducing computational cost.
>
> While ATV requires slightly more memory, the increase is limited in practice. Overall, ATV maintains strong training efficiency while offering improved adaptability over existing parameter-efficient tuning methods.
>
> > **W2)** Why does the paper use the decoder-only GPT family models? How does a decoder-only model generate a vector representation? Is it using its last layer hidden state?
>
> Thank you for raising this important point. We agree that encoder-based models are intuitive choices for generating query representations. In our main experiments, we use GPT-2 primarily to match the decoder-only architecture of the target model (Llama-3).
>
> To address this suggestion, we additionally evaluate ATV using encoder-based generators (BERT and Sentence-Transformer) while keeping the target model fixed.
>
> For BERT, we use the final-layer hidden state of the [CLS] token, which is standard for sentence-level representations. For Sentence-Transformer, we apply mean pooling, following its objective of producing semantically meaningful sentence embeddings.
>
> | Generator (for ATV)  | NLU | Reasoning | Knowledge | Math | Safety | Avg. |
> | -------------------- | -------------: | -------------: | -------------: | -------------: | -------------: | -------------: |
> | BERT | **61.4 ± 4.7** | 74.5 ± 0.6 | 70.8 ± 1.6 | 24.1 ± 2.9 | 72.6 ± 3.7 | 60.7 ± 0.4 |
> | Sentence-Transformer | 60.1 ± 0.8 | 75.5 ± 1.5 | 70.9 ± 0.3 | 25.0 ± 1.0 | 72.8 ± 2.2 | 60.9 ± 0.4 |
> | GPT-2 | 61.0 ± 5.0 | **76.1 ± 1.3** | **73.0 ± 1.6** | **25.8 ± 2.0** | **74.8 ± 0.4** | **62.1 ± 1.5** |
>
> While all generators yield competitive results, GPT-2 achieves the highest overall accuracy. This outcome likely reflects a closer representational compatibility between the generator and the decoder-style target model. Prior work shows that transfer methods are generally more effective when the source and target models share similar architectures or training paradigms, whereas transferring across heterogeneous PLM families often introduces misaligned embedding spaces that require additional projection or alignment mechanisms to compensate for structural differences [1, 2].
>
> In our setting, GPT-2 benefits from this architectural match, leading to more stable task-vector formation and more reliable downstream steering of the target model. We have incorporated these additional results into the revised manuscript for completeness.

---

> > ### Author Response · Authors · 2025-11-28
> >
> > > **Q1)** Does ATV require access to ICL demonstrations? For traditional TV methods, they typically extract TVs from a set of ICL demonstrations. It seems that ATV here does not require this, but only tasks for training.
> >
> > Our method does not require ICL-style demonstrations as used in traditional task-vector approaches. During training, ATV relies on supervised (x,y) pairs, where x is the input query and y is the corresponding target output. These pairs are used in a standard supervised-learning manner rather than being formatted as in-context demonstrations or inserted into prompts to extract task vectors.
> >
> > At inference time, ATV operates without demonstrations. The system generates a query-conditioned task vector based solely on the input query.
> >
> > Overall, ATV uses supervised (x,y) data during training but does not employ ICL demonstrations in the conventional sense either at training or evaluation time. This represents a conceptual distinction from prior task-vector methods that explicitly derive task vectors from demonstration prompts.
> >
> > > **Q2)** For tasks where output is a phrase or sentence (i.e., y is not a single token), how is the CE loss in Eq. 4 computed?
> >
> > For tasks whose target output consists of multiple tokens, the cross-entropy loss in Eq. 4 is computed over the entire sequence by taking the mean of the token-level cross-entropy across all decoding steps. To verify that ATV functions as intended under such multi-token conditions, we evaluate the method on GSM8K, a dataset requiring multi-step and multi-token reasoning. We compared single-token injection, where the task vector is applied only at the first decoding step, with multi-token injection, where it is applied throughout the generation process. The results are presented below.
> >
> > | Method | Accuracy   |
> > | ---------------------------- | ---------- |
> > | Zero-shot | 30.3 ± 1.5 |
> > | ATV (Single-token injection) | 35.3 ± 4.0 |
> > | ATV (Multi-token injection)  | 47.0 ± 2.6 |
> >
> > The multi-token injection setting achieves higher accuracy and lower variance than the single-token configuration. These results confirm that the sequence-level cross-entropy formulation in Eq. 4 is appropriate for multi-token outputs and that the ATV mechanism behaves as intended in such settings. These results and the accompanying analysis have been added to the revised manuscript.
> >
> > We sincerely appreciate the reviewer’s insightful questions and constructive feedback. Your comments have been invaluable in helping us clarify important aspects of our approach and strengthen the overall presentation of the work. We believe the revisions based on your suggestions have meaningfully improved the quality of the paper.
> >
> > ---
> >
> > [1] Su, Yusheng, et al. "On transferability of prompt tuning for natural language processing." Proceedings of the 2022 Conference of the North American Chapter of the Association for Computational Linguistics: Human Language Technologies. 2022.
> >
> > [2] Wu, Zijun, Yongkang Wu, and Lili Mou. "Zero-Shot Continuous Prompt Transfer: Generalizing Task Semantics Across Language Models." The Twelfth International Conference on Learning Representations (2024).

---

> > > ### Author Response · Authors · 2025-12-03
> > >
> > > We appreciate the reviewer’s constructive comments. In response, we conducted a detailed examination of the training cost associated with the additional modules, added evaluations with encoder-based generators to clarify the architectural choice, and further analyzed multi-token settings to address sequence-level behavior. We would be pleased to provide any additional clarification or continue the discussion.

---

### Official Review · Reviewer_qfNR · 2025-11-01

**Soundness:** 2
**Presentation:** 2
**Contribution:** 2
**Rating:** 4
**Confidence:** 4

**Summary:**

The paper proposes Adaptive Task Vectors (ATV), a method for steering frozen LLMs by injecting a query-conditioned vector into selected layers. A small generator network produces this vector from the input query, which is then added to hidden states to modulate predictions.

**Strengths:**

- Simple and computationally light steering mechanism for frozen LLMs.
- Clear motivation to reduce inference-time token overhead compared to ICL.
- Interesting empirical finding that early-layer injection performs best.
- Theoretical framing situates ATV among PEFT methods (LoRA, prefix, prompt tuning).

**Weaknesses:**

- I find the framing of the method somewhat misaligned. Despite its name, the proposed vector is generated per query via supervised training and is not shared across examples or tasks. This makes it a query-specific steering signal rather than a reusable task-level representation. A more appropriate baseline would embed fixed demonstrations once into a shared vector that can be reused across queries, achieving comparable token efficiency while preserving task conditioning. Moreover, the notion of “adaptive” task vectors feels conceptually weak: most token overhead in ICL stems from contextual demonstrations, not the query itself, so producing a query-conditioned vector offers limited efficiency gains in practice.
- The generator is trained with labeled data but receives only the query text at test time.
  It is unclear how such a setup supports genuine task-level generalization rather than multitask supervised fitting.
- The claim that early-layer injection aligns with prior findings on layer specialization is questionable.
  If lower layers primarily encode lexical features, injecting task information there should not yield optimal results.
  In my opinion, a more likely explanation is residual bias shaping rather than the injection of semantic "task meaning."
  Quantitative analyses could clarify this mechanism.
- Table 2 sourcing -- it is unclear whether results "from ELICIT" were reproduced or copied directly, raising reproducibility concerns.
- The reported standard deviations for ATV are large, yet no statistical tests are provided. This brings into question the significance of reported improvements.
- The paper claims that ATV achieves greater efficiency by requiring fewer input tokens than prompt-based ICL. While this is valid in principle, similar token savings could likely be achieved by amortizing demonstration information, for instance, encoding a fixed set of $k$ demonstrations once into a shared vector reused across queries, while still preserving explicit task conditioning.
- Theorem 1 analyzes an affine "static ATV" that includes a multiplicative low-rank term  $\Delta W_\ell(v) h_\ell,$ whereas the implemented ATV is purely additive. If I understood it correctly, the equivalence result therefore applies to a broader, unimplemented variant rather than to the actual method. Moreover, the proof treats hidden states $h_\ell$ as free variables instead of $h_\ell = f_\ell(x)$, so the result holds in hidden-state space but not necessarily for real input mappings.

**Questions:**

1. Can you conceptually explain how does the generator generalize to unseen tasks without demonstrations or label schemas?
2. Why should early-layer injection be optimal if these layers are mainly lexical?

---

> ### Author Response · Authors · 2025-11-28
>
> Thank you for your careful and constructive review. We sincerely appreciate the time and thought you devoted to evaluating our submission. Your comments were valuable in helping us clarify several conceptual points, strengthen our empirical justification, and better articulate the intended scope of the proposed framework.
>
> Below, we respond to each of your concerns in detail and aim to address the issues you raised as clearly as possible.
>
> > **W1)** I find the framing of the method somewhat misaligned.
>
> Prior work typically defines a “task vector” as a static, task-level representation. Our method preserves this general formulation but extends it by generating task vectors conditioned on each query, representing a natural dynamic variant of the task vector framework.
>
> For baselines, methods such as ELICIT and I2CL already represent the shared-vector setting described in the comment since they rely on vectors constructed from fixed demonstrations. In our experiments, these static approaches showed clear limitations in OOD and adversarial evaluations such as HANS, whereas ATV remained stable. This supports the need for query-conditioned vectors.
>
> In terms of computational cost, ATV introduces only minor overhead, as its generator is a small language model and its expansion module consists of a single linear layer. As shown in Section 4.6, its inference time is comparable to LoRA and I2CL, yet lower than retrieval-based methods such as ELICIT. Despite the comparable cost, ATV achieves substantially higher accuracy than static task vector baselines.
>
> > **W2)** It is unclear how such a setup supports genuine task-level generalization rather than multitask supervised fitting.
>
> The small generator (GPT-2) does not perform the task nor generate outputs. The frozen LLM remains fully responsible for generation, whereas GPT-2 functions solely as an auxiliary module that interprets the query and produces a small Δ that is added to the hidden representation. The role of this Δ is to provide a lightweight, query-dependent adjustment rather than to solve the task itself.
>
> Recent work indicates that such low-rank or auxiliary Δ updates can strengthen robustness. PETL (Parameter efficient transfer learning) studies report that LoRA-style modules often improve OOD behavior compared to full fine-tuning [1], and SeTAR shows that selective low-rank updates alone can enhance OOD detection in frozen models [2]. These observations suggest that a query-conditioned Δ can enhance the frozen LLM's ability to generalize to unseen tasks.
>
> > **W3)** The claim that early-layer injection aligns with prior findings on layer specialization is questionable.
>
> Thank you for this insightful comment. We find this perspective particularly helpful, especially the point regarding residual bias shaping as a more plausible explanation for the observed effect.
>
> This suggestion prompts us to revisit our initial interpretation, and we agree that this framing offers a clearer understanding of the underlying mechanism. We have incorporated this clarification into the revised discussion of the manuscript.
>
> > **W4)** Table 2 sourcing; it is unclear whether results "from ELICIT" were reproduced or copied directly, raising reproducibility concerns.
>
> The ELICIT numbers in Table 2 are taken directly from the original ELICIT paper rather than reproduced on our side. We have clarified this in the manuscript and explicitly marked the source to avoid confusion. We appreciate this comment and agree that transparent sourcing is important for reproducibility.
>
> > **W5)** The reported standard deviations for ATV are large, yet no statistical tests are provided.
>
> To address this concern, we conduct paired t-tests using the results from all tasks in both the in-domain and unseen settings. For each model, we compared ATV against the strongest-performing baseline, BM25. The p-values are reported in the table below and fall below standard significance thresholds in most cases.
>
> | Model    | ID ATV | ID Best Baseline | ID p-value | Unseen ATV | Unseen Best Baseline | Unseen p-value |
> |----------|-------|------------------|------------|----------------|----------------|--------------|
> | Llama-3  | 62.1 ± 1.5 | 55.8 ± 0.7 | **0.0106** | 63.4 ± 2.5 | 56.4 ± 0.4 | 0.0742 |
> | Mistral  | 58.3 ± 1.3 | 55.5 ± 0.4 | 0.0562     | 59.4 ± 2.6 | 49.2 ± 0.3 | **0.0375** |
>
> For example, Llama-3 yields a p-value of 0.0106 in the in-domain setting, and Mistral yields 0.0375 in the unseen setting, indicating that the improvements achieved by ATV are statistically significant despite the larger standard deviations. We have incorporated these results into the revised manuscript to ensure clarity and completeness. We appreciate this comment, which prompted us to verify and report the statistical significance of our findings more rigorously.

---

> > ### Author Response · Authors · 2025-11-28
> >
> > > **W6)** The paper claims that ATV achieves greater efficiency by requiring fewer input tokens than prompt-based ICL. While this is valid in principle, similar token savings could likely be achieved by amortizing demonstration information.
> >
> > While token savings can indeed be achieved by constructing a shared vector from fixed demonstrations and reusing it across queries, our results show that such static representations are fundamentally limited in robustness. As shown in our OOD and adversarial evaluations, particularly on HANS, methods relying on a single shared or static vector, including ELICIT-style retrieval of fixed task representations, degrade substantially under distribution shift.
> >
> > In contrast, ATV maintains stable performance. These results indicate that the key distinction lies not in token efficiency alone, but in the ability of a query-conditioned Δ to support generalization beyond the training distribution.
> >
> > > **W7)** Theorem 1 analyzes an affine "static ATV" that includes a multiplicative low-rank term, whereas the implemented ATV is purely additive.
> >
> > We appreciate this thoughtful comment and address the two concerns below.
> >
> > **(1) Regarding the use of a more general “static ATV” in Theorem 1:**
> > Theorem 1 compares the expressive class of rank-$r$ update operators in ATV and LoRA. The implemented additive ATV corresponds to the $r$ = 0 instance of this general formulation. Therefore, the additive version used in our experiments is included as a special case of the theorem, and its expressive equivalence follows directly from the general result.
> >
> > **(2) Regarding the treatment of hidden states as free variables:**
> > The proof constructs update operators $\Delta W_\ell$ and biases $\beta_\ell$ that are parameter-fixed and input-independent. For any input sequence $x$, both models therefore apply identical layer-wise transformations under the same $\Delta W_\ell$ and $\beta_\ell$. A forward induction then shows that the final hidden state $h^{T}{L}(x)$, and consequently the next-token distribution $F(x)$, match for every $x$. Replacing the symbolic $h\ell$ with the concrete $h_\ell(x)$ therefore leaves the argument intact, as the equivalence holds for all inputs.
> >
> > In summary, (1) the additive ATV implemented in our experiments is covered as the $r$ = 0 special case of Theorem 1, and (2) the equivalence continues to hold when hidden states are instantiated as $h_\ell(x)$, since the comparison concerns input-independent operators whose action remains identical for every input sequence.
> >
> > We thank the reviewer again for raising these points and hope that the clarification above addresses the concern. We have also incorporated a concise explanation of these details into the revised manuscript for completeness.
> >
> > > **Q1)** Can you conceptually explain how does the generator generalize to unseen tasks without demonstrations or label schemas?
> >
> > As noted in our response to W2, the generator does not perform the task itself. Its role is to produce a small query-conditioned Δ that modulates the frozen LLM’s hidden representations. Because this Δ reflects structural signals present in the input, the mechanism naturally generalizes to unseen tasks, which is consistent with the OOD behavior observed in our experiments.
> >
> > > **Q2)** Why should early-layer injection be optimal if these layers are mainly lexical?
> >
> > As noted in our response to W3, we find this interpretation reasonable and agree that the residual-bias perspective provides a more appropriate explanation than assuming direct semantic injection into early layers. We have revised the manuscript accordingly.
> >
> > Thank you again for your careful review and constructive feedback. We believe the revisions and clarifications above address the concerns raised, and we appreciate the opportunity to improve the manuscript.
> >
> > ---
> >
> > [1] Cho, Hyunsoo, et al. "Probing out-of-distribution robustness of language models with parameter-efficient transfer learning." Proceedings of the 12th Joint Conference on Lexical and Computational Semantics (* SEM 2023). 2023.
> >
> > [2] Li, Yixia, et al. "Setar: Out-of-distribution detection with selective low-rank approximation." Advances in Neural Information Processing Systems 37 (2024): 72840-72871.

---

> ### Author Response · Authors · 2025-12-03
>
> We thank the reviewer for the thoughtful feedback. In the revisions, we validated the effectiveness of the query-conditioned update over static baselines through empirical comparisons, incorporated the reviewer’s perspective on early-layer effects, and strengthened the statistical and theoretical grounding. We believe these updates fully address the raised concerns and remain available for further discussion.

---

### Official Review · Reviewer_YGMn · 2025-11-01

**Soundness:** 3
**Presentation:** 3
**Contribution:** 2
**Rating:** 6
**Confidence:** 3

**Summary:**

The paper proposes Adaptive Task Vectors (ATV), which use a small generator model to produce per-input vectors that are expanded and injected additively into the last-token hidden states across layers of a frozen LLM, aiming to preserve ICL’s flexibility while avoiding prompt-token overhead and fixed-vector rigidity. Theoretically, ATV is argued to match LoRA’s expressivity under equal rank budgets and to strictly subsume Prefix-Tuning under a linear attention approximation.

**Strengths:**

- Clear articulation of a simple, modular pipeline that keeps the target LLM frozen while learning a lightweight generator and linear expansion, with a consistent injection interface across layers.
- Theoretical framing that precisely scope-limits claims to next-token distributions and gives a clean equivalence-to-LoRA result under matched rank and placements, plus a principled argument for subsuming Prefix-Tuning under a linearized attention view.
- Broad empirical sweep over a standardized ELICIT setup with in-domain and held-out tasks, plus adversarial HANS evaluation and ablations on generator size and injection depth, offering multiple angles on behavior and efficiency.

**Weaknesses:**

- The LoRA equivalence and superiority-over-Prefix claims rest on constrained scopes and approximations: the LoRA equivalence focuses only on next-token distribution with matched placements and static ATV, and the Prefix result relies on a linear attention approximation that may diverge from real softmax attention in practice.
- The empirical fairness of baselines is questionable in places. For example, the LoRA setup departs from the original learning rate due to poor performance (adjusted to 4e-4), and ELICIT requires selecting an optimal injection layer per task, which may advantage or disadvantage baselines depending on protocol specifics not fully stress-tested across settings.
- ATV underperforms BM25 retrieval on math, and while the paper attributes this to pattern alignment and shows targeted math-only training helps, this suggests ATV’s generality depends on careful domain data allocation, weakening one-size-fits-all claims across reasoning vs. procedural domains.
- The dynamic generator introduces an extra forward pass and a large linear expansion into RLdl parameters per input; while inference latency is reported comparable to LoRA, the memory/runtime implications of full-layer injection and expansion mapping are undercharacterized for larger models and batched, long-sequence workloads.
- Injection is additively applied to the last-token hidden state per layer, which may limit control over earlier token positions and long-context interactions; the paper lacks analysis of multi-token or position-aware injection variants and their trade-offs.
- Claims of generalization could be confounded by the close coupling to the ELICIT evaluation harness and prompt templates; robustness to substantially different prompting regimes, decoding strategies, or safety constraints is not deeply probed beyond format adherence and limited consistency metrics.
- Safety and bias implications of steering hidden states are only cursorily addressed via a few safety datasets; there is no analysis of potential exacerbation or mitigation of harmful behaviors when the generator produces misaligned vectors for out-of-distribution inputs.

**Questions:**

- How sensitive is ATV to the exact placement and breadth of layer injections. Could partial-layer injection with structured sparsity retain most gains while cutting expansion cost and minimizing interference?
- Can the authors quantify generator–target mismatch effects (e.g., cross-family or multilingual generalization) and whether external encoders or distilled generators alter robustness vs. efficiency trade-offs?
- How does ATV interact with decoding strategies like temperature, nucleus sampling, or constrained decoding—does the steering effect persist uniformly or require retuning?
- Could multi-token or segment-level injections improve math/procedural tasks and adversarial robustness, and what are the stability risks when modifying more than the last-token state?

---

> ### Author Response · Authors · 2025-11-28
>
> We sincerely thank the reviewer for the thoughtful and constructive feedback.
> Your comments were highly valuable, and they helped us better understand the key concerns and areas requiring clarification.
> Below, we address each of the points you raised in detail.
>
> > **W1)** The LoRA equivalence and superiority-over-Prefix claims rest on constrained scopes and approximations.
>
> Our LoRA equivalence result is an exact statement: when the rank budget and insertion placements are matched, static ATV and LoRA generate the same set of next-token distributions $F(x)$ for every input sequence. The proof treats the transformer’s attention mechanism as a black box and does not approximate or linearize the softmax kernel in any way. Consequently, the theorem remains valid regardless of the specific attention form upstream of the hidden states.
>
> For the Prefix-Tuning comparison, we intentionally adopt the same analytical abstraction used in the original Prefix-Tuning paper [1], combining next-token analysis with a linearized attention surrogate. This shared framework, which is also used in subsequent work such as Performer [2], enables a clean, like-for-like comparison of structural capacity.
>
> Although our theoretical analysis focuses on next-token prediction, this objective is not an approximation but the exact quantity optimized and evaluated by modern language models. Within this standard formulation, our results accurately capture the structural differences between ATV and existing methods. Furthermore, the empirical results exhibit trends consistent with the theoretical analysis, indicating that the conclusions remain informative even under these simplified assumptions.
>
> > **W2)** The empirical fairness of baselines is questionable in places.
>
> We clarify our baseline choices for LoRA and ELICIT below.
>
> **1) LoRA learning rate.**
> Using the original learning rate of 1e-3 resulted in the following performance in our setting:
>
> | Setting | ID | OOD |
> | ------------------------ | ------ | ----- |
> | LoRA (1e-3, original LR) | 32.094 | 19.23 |
>
> Because this performance was substantially lower than that of other baselines, we adjusted the learning rate to 4e-4 so that LoRA could function properly within our experimental environment. We apologize for any confusion this change may have caused.
>
> **2) ELICIT layer selection.**
> ELICIT’s methodology includes selecting an optimal injection layer per task, and we followed this protocol to reproduce the method faithfully. For completeness, we also report results for injection into all layers (Section 4.5, Table 5), using the same configuration as ATV and I2CL.
>
> The results in that table show that ATV outperforms ELICIT even without any layer selection, indicating that our conclusions are not sensitive to this aspect of the baseline.
>
> > **W3)** ATV underperforms BM25 retrieval on math.
>
> We acknowledge that ATV underperforms BM25 on Math tasks in our evaluation. The Math dataset we use follows a single-token generation format, where the model must map complex multi-step reasoning to a single final output token. In such cases, controlling the model using only a single task vector may be insufficient to capture the full reasoning path required for accurate prediction.
>
> We view this as a valuable direction for extending ATV toward more expressive task representations, particularly for reasoning-oriented tasks. We appreciate the reviewer’s insight and believe this points to a meaningful avenue for future work.

---

> > ### Author Response · Authors · 2025-11-28
> >
> > > **W4)** The dynamic generator introduces an extra forward pass and a large linear expansion into RLdl parameters per input.
> >
> > We evaluated the runtime and memory characteristics of ATV using a large model (Llama-2-13B) and measured both inference time and peak memory across datasets with substantially different prompt lengths. HellaSwag contains the longest prompts in our evaluation, while BBH Boolean contains the shortest, allowing us to assess sensitivity to sequence length.
> > sec/sample denotes the end-to-end inference time for one prompt instance.
> >
> > **Inference Time (Llama-2-13B).**
> >
> > | Method | Inference Time (sec/sample) | HellaSwag (Longest) | BBH Boolean (Shortest) | Accuracy |
> > | ------ | ------------------ | ------------------- | ---------------------- | ------------------ |
> > | I2CL   | 0.0448 | 0.0556 | 0.0428 | 40.9 ± 0.7 |
> > | ELICIT | 0.1390 | 0.2349 | 0.0676 | 34.0 ± 0.0 |
> > | ATV    | 0.0555 | 0.0667 | 0.0527 | 57.7 ± 0.7 |
> >
> > ATV shows only a modest increase relative to I2CL and remains stable even on HellaSwag, the longest-prompt dataset. By contrast, ELICIT exhibits substantially higher latency across all settings.
> >
> > **Peak Memory (Llama-3-8B).**
> >
> > | Method | Inference Time (sec/sample) | Peak Memory (MB) |
> > | ------ | --------------------------- | ----------------- |
> > | LoRA   | 0.030 | 15822.2 |
> > | I2CL   | 0.037 | 15316.5 |
> > | ATV    | 0.047 | 16190.0 |
> > | ELICIT | 0.077 | 15896.2 |
> >
> > ATV’s peak memory usage is slightly higher than I2CL but comparable to ELICIT, and no substantial increases are observed in practice.
> >
> > Overall, these measurements indicate that ATV’s generator and expansion modules introduce minimal runtime and memory overhead, even under long-prompt conditions.
> >
> > > **W5)** The paper lacks analysis of multi-token or position-aware injection variants and their trade-offs.
> >
> > To explore whether multi-token or segment-level injections offer advantages for procedural or math-intensive tasks, we conducted an additional experiment on GSM8K, a dataset that requires multi-step, multi-token reasoning. We compared (i) single-token injection, applied only at the first decoding step, and (ii) multi-token injection, applied throughout the generation process. The results are summarized below.
> >
> > | Method | Accuracy   |
> > | ---------------------- | ---------- |
> > | Zero-shot | 30.3 ± 1.5 |
> > | ATV (Single-token injection) | 35.3 ± 4.0 |
> > | ATV (Multi-token injection)  | 47.0 ± 2.6 |
> >
> > The multi-token injection setting shows a noticeable improvement and reduced variance relative to the single-token variant. This suggests that, for tasks requiring multi-step reasoning, maintaining task information across decoding steps can be beneficial. We also observed no instability when modifying more than the last-token state, indicating that multi-token or segment-level injection is a viable extension of ATV for procedural tasks.
> >
> >
> > > **W6)** Claims of generalization could be confounded by the close coupling to the ELICIT evaluation harness and prompt templates; robustness to different prompting regimes, decoding strategies, or safety constraints is not deeply probed.
> >
> > To assess the robustness of ATV under different decoding and prompting regimes, we conduct an additional evaluation on GSM8K using the same multi-token injection setup described in W5. While the main experiments relied on greedy decoding, this analysis varied the decoding configuration by adjusting the temperature (0.3 and 0.7) and by applying nucleus sampling. The results are presented below.
> >
> > | Decoding Strategy | Zero-Shot Accuracy | ATV Accuracy |
> > |-------------------|------------------|-------------------|
> > | Greedy | 30.3 ± 1.5 | 47.0 ± 2.6 |
> > | Temp = 0.3 | 28.7 ± 4.0 | 46.3 ± 5.1 |
> > | Temp = 0.7 | 21.7 ± 4.7 | 38.3 ± 6.4 |
> > | Nucleus | 15.3 ± 1.6 | 38.7 ± 6.7 |
> >
> > Both Zero-Shot and ATV exhibit some variability as the decoding strategy changes. However, the degradation is consistently smaller for ATV, and its accuracy remains higher across all evaluated configurations. This indicates that the steering effect of ATV does not rely heavily on a specific decoding procedure.
> >
> > The main paper also evaluates ATV under training and test settings with different answer templates. ATV maintains strong performance despite this mismatch, further suggesting robustness to prompt-level variation and indicating that the method is not tightly coupled to a specific prompting format.

---

> > > ### Author Response · Authors · 2025-11-28
> > >
> > > > **W7)** Safety and bias implications of steering hidden states are only cursorily addressed via a few safety datasets.
> > >
> > > We acknowledge that manipulating hidden states can introduce safety and bias risks if the injected vector is misaligned.
> > >
> > > To assess this risk, we evaluate ATV not only on standard safety datasets but also on HANS, which probes whether a model relies on superficial heuristics or maintains robust decision boundaries. ATV showed stable performance on HANS, suggesting that the task vector does not distort the model’s underlying representations excessively and instead preserves its original decision structure while guiding it toward the intended task.
> > >
> > > While broader safety analysis remains an important direction for future work, our results suggest that ATV maintains robust behavior, contributing to its overall stability in practice.
> > >
> > > > **Q1)** How sensitive is ATV to the exact placement and breadth of layer injections?
> > >
> > > We assess the sensitivity of ATV to the placement and breadth of layer injection, and the results are reported in Section 4.5, Table 5. In addition to the full-layer configuration, we evaluated layer-wise settings that inject the task vector into only a subset of layers.
> > >
> > > Notably, injecting only into the bottom layer leads to minimal performance degradation relative to the full-layer version. This indicates that ATV does not strictly rely on injecting into all layers and can operate effectively with a reduced injection scope.
> > >
> > > > **Q2)** Can the authors quantify generator–target mismatch effects and whether external encoders or distilled generators alter robustness vs. efficiency trade-offs?
> > >
> > > To examine generator–target mismatch, we evaluate ATV using generators from different model families while keeping the target model fixed as a decoder-only Llama-3. Specifically, we compare two encoder-based generators (BERT, Sentence-Transformer) with the decoder-based GPT-2 used in our main experiments. The results are provided below.
> > >
> > > | Generator (for ATV)  |        NLU |  Reasoning |  Knowledge |       Math |     Safety |       Avg. |
> > > | -------------------- | ---------: | ---------: | ---------: | ---------: | ---------: | ---------: |
> > > | BERT                 | **61.4 ± 4.7** | 74.5 ± 0.6 | 70.8 ± 1.6 | 24.1 ± 2.9 | 72.6 ± 3.7 | 60.7 ± 0.4 |
> > > | Sentence-Transformer | 60.1 ± 0.8 | 75.5 ± 1.5 | 70.9 ± 0.3 | 25.0 ± 1.0 | 72.8 ± 2.2 | 60.9 ± 0.4 |
> > > | GPT-2                | 61.0 ± 5.0 | **76.1 ± 1.3** | **73.0 ± 1.6** | **25.8 ± 2.0** | **74.8 ± 0.4** | **62.1 ± 1.5** |
> > >
> > > GPT-2 achieves the highest overall performance among the three generators. This pattern aligns with prior findings showing that transfer methods are most effective when the source and target models share similar architectures or training paradigms, while transfer across different PLM series often requires separate projection or alignment mechanisms due to mismatched embedding spaces [3, 4].
> > >
> > > These results indicate that ATV operates effectively with heterogeneous generators, although using one that matches the target model’s backbone, as in the GPT-2 setting, yields the most stable and effective transfer.
> > >
> > > > **Q3)** How does ATV interact with decoding strategies like temperature, nucleus sampling, or constrained decoding—does the steering effect persist uniformly or require retuning?
> > >
> > > To assess how ATV interacts with different decoding strategies, we analyze the GSM8K results reported in W6, where temperature-based decoding and nucleus sampling were applied alongside greedy decoding. Across all configurations, ATV consistently achieved higher accuracy than the zero-shot baseline and exhibited smaller performance degradation as the decoding strategy changed. These findings indicate that the steering effect of ATV remains stable under standard variations in decoding parameters and does not depend on a specific decoding procedure.

---

> > > > ### Author Response · Authors · 2025-11-28
> > > >
> > > > > **Q4)** Could multi-token or segment-level injections improve math/procedural tasks and adversarial robustness, and what are the stability risks when modifying more than the last-token state?
> > > >
> > > > Following the analysis in W5, we evaluate multi-token generation on GSM8K to assess the impact of extending injection beyond the final hidden state in a multi-step reasoning context. GSM8K requires generating a sequence of intermediate reasoning steps, and injecting the task vector throughout the decoding process produced higher accuracy and lower variance than single-token injection.
> > > >
> > > > These results indicate that, within the GSM8K setting, multi-token or segment-level injection can be applied in a controlled manner and may further enhance performance on tasks requiring extended reasoning.
> > > >
> > > > We thank the reviewer for the careful assessment and constructive remarks. The points raised have been addressed in the revisions, and we believe the manuscript is now substantially clearer and more rigorous.
> > > >
> > > > ---
> > > >
> > > > [1] Li, Xiang Lisa, and Percy Liang. "Prefix-tuning: Optimizing continuous prompts for generation." arXiv preprint arXiv:2101.00190 (2021).
> > > >
> > > > [2] Choromanski, Krzysztof, et al. "Rethinking attention with performers." arXiv preprint arXiv:2009.14794 (2020).
> > > >
> > > > [3] Su, Yusheng, et al. "On transferability of prompt tuning for natural language processing." Proceedings of the 2022 Conference of the North American Chapter of the Association for Computational Linguistics: Human Language Technologies. 2022.
> > > >
> > > > [4] Wu, Zijun, Yongkang Wu, and Lili Mou. "Zero-Shot Continuous Prompt Transfer: Generalizing Task Semantics Across Language Models." The Twelfth International Conference on Learning Representations (2024).

---

> > > > > ### Author Response · Authors · 2025-12-03
> > > > >
> > > > > We once again appreciate the reviewer’s careful assessment. The revisions include clearer theoretical clarification, expanded efficiency evaluation, additional experiments on reasoning-oriented tasks, an extended analysis of multi-token injection, and further evaluations using encoder-based generators to address architectural considerations. We believe these updates substantively engage with the primary concerns raised in the review, and we remain available for further discussion should the review process allow.

---

### Author Response · Authors · 2025-11-28
**General Response**

We thank all four reviewers (`YGMn`, `qfNR`, `Hwr1`, and `LSEb`) for their detailed and constructive feedback. This general response summarizes the main strengths and key concerns raised in the reviews, and explains how the revisions and additional analyses address them. Reviewer-specific clarifications are provided in the individual responses.

**Strengths highlighted by reviewers:**

* **[`YGMn`, `Hwr1`]** Clear and modular design of Adaptive Task Vectors (ATV), enabling adaptive control of frozen LLMs through a lightweight generator.
* **[`YGMn`, `qfNR`]** Well-scoped theoretical formulation, including equivalence-to-LoRA and discussion of Prefix-Tuning under linearized attention.
* **[`Hwr1`, `LSEb`]** Broad empirical evaluation covering in-domain, unseen, and adversarial tasks, with ablations on generator, layer placement, and efficiency.
* **[`LSEb`, `YGMn`]** Organized presentation and reproducible experimental setup supported by clear figures and tables.

**Concerns and how we addressed them:**

* **Scope of theoretical results (`YGMn`, `qfNR`):** We clarified the assumptions behind the LoRA equivalence and Prefix-Tuning analyses and aligned them with standard formulations used in prior work.
* **Baseline fairness and reproducibility (`YGMn`, `qfNR`):** Baseline configurations, learning rate adjustments, and dataset sources are now explicitly documented; results remain consistent under alternative setups.
* **Motivation and query-conditioned design (`LSEb`, `qfNR`):** New ablations comparing static and input-conditioned variants show that conditioning on each query significantly improves generalization. Specifically, static baselines collapsed on the OOD benchmark (HANS), whereas ATV maintained robust performance, validating the necessity of query-specific modulation.
* **Generator design (`Hwr1`, `LSEb`):** Additional experiments with encoder-based generators (BERT, Sentence-Transformer) and early-layer representations confirm that a small decoder-based generator provides the best performance due to better architectural alignment with the target LLM, while remaining computationally efficient.
* **Multi-token injection (`YGMn`, `Hwr1`):** New experiments on GSM8K demonstrate that applying the vector across multiple decoding steps improves performance without instability, addressing concerns about procedural and reasoning tasks.
* **Data scale and statistical validation (`LSEb`, `qfNR`):** Experiments with 16, 90, and 256 samples and paired t-tests across tasks verify that ATV’s gains are consistent and statistically significant.
* **Efficiency and runtime (`YGMn`, `Hwr1`):** Detailed measurements show ATV maintains training and inference cost comparable to LoRA and I2CL, with negligible additional memory usage.
* **Additional clarifications (`LSEb`):** The revision adds λ-sensitivity analysis, missing references (Todd et al., 2024; Saglam et al., 2025), and extended comparison with static task-vector baselines.

We again thank all reviewers for their constructive insights. The feedback has improved both the clarity and completeness of the work, and we will continue refining the manuscript during the rebuttal period.

---

### Author Response · Authors · 2025-12-03
**[To New AC] Summary of Rebuttal Outcome**

**Dear Area Chair,**

We understand that you have been newly assigned to our submission due to the recent incident, and that all reviews and scores have reverted to their initial state. We appreciate your effort in taking on this responsibility under an unexpected circumstance.
We respectfully request that you consider the following context as you assess the submission.

**1. Summary of Reviewer Concerns and Our Resolutions**

Below we summarize each reviewer’s primary concerns and the corresponding resolutions we provided in the rebuttal. For more details, please read our whole official comments.

* **Reviewer YGMn (Score: 6)**

    * **Initial concerns:**
    The reviewer questioned the assumptions underlying the theoretical analyses, raised concerns about baseline fairness, and requested analyses on decoding strategies, multi-token injection, and generator–target mismatch.
    * **Resolution:**
    In response, we refined the theoretical explanations, clarified the underlying assumptions, and documented baseline configurations. We also conducted experiments demonstrating ATV’s robustness across diverse decoding strategies and analyzed the effects of multi-token injection on GSM8K. In addition, we included generator ablations to assess potential mismatch effects. These revisions address the reviewer’s concerns regarding analytic scope, robustness, and the fairness of the experimental comparisons.

* **Reviewer qfNR (Score: 4)**

    * **Initial concerns:**
    The reviewer challenged the conceptual framing of query-conditioned vectors, raised concerns about statistical rigor, and questioned distinctions from static task-vector methods.
    * **Resolution:**
    We provided ablations comparing static and dynamic vector variants, showing that input-conditioned vectors yield significantly more robust behavior. We also added paired t-tests validating that ATV’s improvements over the best baseline (BM25) are statistically significant. These analyses directly address questions regarding the necessity and effect of query conditioning.

* **Reviewer Hwr1 (Score: 6)**

    * **Initial concerns:**
    The reviewer asked about the training cost of the added modules, the choice of a decoder-based generator, and the handling of multi-token outputs.
    * **Resolution:**
    We provided training-time and memory comparisons showing that the added modules remain efficient. We included generator ablations using encoder-based models (BERT and Sentence-Transformer) and explained the architectural reasons for choosing a decoder-style generator. Additionally, we presented GSM8K multi-token experiments confirming that ATV behaves correctly under multi-step decoding scenarios.

* **Reviewer LSEb (Score: 2)**

    * **Initial concerns:**
    This reviewer raised concerns about the motivation for query-conditioned vectors, the necessity of dynamic modulation over static task representations, and the choice of GPT-2 relative to encoder-based alternatives. Further issues included missing references, the absence of lambda sensitivity analysis, and the need for ablations validating the two-stage architecture.
    * **Resolution:**
    We added static-vector baselines including LTV to illustrate the limitations of fixed representations under distribution shift, clarified the functional motivation for query-conditioned modulation. Furthermore, we incorporated the missing references, and provided lambda sensitivity results. We also expanded the empirical analysis with statistical testing, encoder-based generator ablations, and early-layer representation comparisons, offering a comprehensive response to the reviewer’s points.

**2. Key Contributions and Rebuttal Highlights**

The rebuttal period enabled us to substantially strengthen the submission in several respects:

* We expanded our analyses through additional experiments on generator design, multi-token inference, data scaling, and statistical significance testing, and clarified several methodological points raised across reviews.
* We conducted comparative studies showing that static global vectors, per-task vectors, and LTV-based approaches are consistently less robust than the input-conditioned mechanism in ATV, especially in unseen and adversarial settings, empirically validating the motivation for our dynamic design.
* We documented baseline configurations more thoroughly and provided supplemental empirical evidence supporting the efficiency and practicality of the proposed modules.

These additions collectively reinforced both the empirical validity and the conceptual grounding of the proposed method.

**Conclusion**

We believe that the concerns raised in the initial reviews have been thoroughly addressed through the clarifications, analyses, and additional experiments provided in our rebuttal, and we hope that these efforts may be helpful to you as you form your assessment.

Thank you for your time and effort in handling this unusual situation.

Sincerely,

The Authors

---

### Meta-Review · Area_Chair_HRLY · 2026-01-07

**Summary:**

The paper proposes Adaptive Task Vectors (ATV), a framework that uses a lightweight "generator" model (e.g., GPT-2) to produce input-specific vectors. These vectors are then injected into the hidden states of a frozen target LLM to steer its output without the prompt-token overhead of In-Context Learning (ICL) or the rigidity of static task vectors.

The core debate during the rebuttal centered on the necessity of dynamic modulation (query-conditioning) versus static representations, the mathematical rigor of the LoRA-equivalence proofs, and the robustness of the method across multi-step reasoning tasks like GSM8K.

**Reviewer Concerns:**

Addressed by Rebuttal

Static vs. Dynamic Motivation (qfNR, LSEb)

Theoretical Expressivity (YGMn, qfNR)

Multi-token & Reasoning Tasks

Computational Efficiency (YGMn, Hwr1)

Not yet:

Generator-Target Mismatch (YGMn)

Reasoning vs. Retrieval: ATV still underperforms

**Reviewer Scores:**

qfNR might raise score to 5/6

LSEb remains skepticism about the GPT-2 choice and the two-stage design may persist

---

### Decision · Program_Chairs · 2026-01-26

Reject